# Transposable Element Populations Shed Light on the Evolutionary History of Wheat and the Complex Co-Evolution of Autonomous and Non-Autonomous Retrotransposons

*Thomas Wicker,\* Christoph Stritt, Alexandros G. Sotiropoulos, Manuel Poretti, Curtis Pozniak, Sean Walkowiak, Heidrun Gundlach, and Nils Stein*

**Wheat has one of the largest and most repetitive genomes among major crop plants, containing over 85% transposable elements (TEs). TEs populate genomes much in the way that individuals populate ecosystems, diversifying into different lineages, sub-families and sub-populations. The recent availability of high-quality, chromosome-scale genome sequences from ten wheat lines enables a detailed analysis how TEs evolved in allohexaploid wheat, its diploids progenitors, and in various chromosomal haplotype segments. LTR retrotransposon families evolved into distinct sub-populations and sub-families that were active in waves lasting several hundred thousand years. Furthermore, It is shown that different retrotransposon sub-families were active in the three wheat sub-genomes, making them useful markers to study and date polyploidization events and chromosomal rearrangements. Additionally, haplotype-specific TE sub-families are used to characterize chromosomal introgressions in different wheat lines. Additionally, populations of non-autonomous TEs co-evolved over millions of years with their autonomous partners, leading to complex systems with multiple types of autonomous, semi-autonomous and non-autonomous elements. Phylogenetic and TE population analyses revealed the relationships between non-autonomous elements and their mobilizing autonomous partners. TE population analysis provided insights into genome evolution of allohexaploid wheat and genetic diversity of species, and may have implication for future crop breeding.**

## 1. Introduction

The hallmark of transposable elements (TEs) is their ability copy themselves and integrate again elsewhere in the genome. Class 1 elements (retrotransposons) replicate primarily via reverse transcription, while Class 2 elements (DNA transposons) have multiple proposed mechanisms of replication, such as replicative transposition or excision site repair using sister chromatids.[1] Independent of the replication mechanism, TEs proliferate much like haploid, asexually reproducing organisms, thereby producing "clonal" lineages that diversify into different sub-families. Although, occasional recombination may occur, for example through template switching during replication,[2,3] TEs mostly proliferate in a clonal manner. Thus, the entirety of a TE family in a given genome much resembles a population of individuals where each TE copy represents an individual. Unlike in populations of organisms, the parental TE copies, however, remain preserved in the genome, thus leaving a "fossil record". This record reaches back only a limited time, because TE-driven genome expansion is counteracted by continuous, and largely random, deletions of DNA so that sequences that are not under selection will eventually be los.[4] In plants, this "genomic turnover" leads to removal of practically all elements that are older than a few million years.[5–8]

The repetitive fraction of plant genomes is dominated by long terminal repeat (LTR) retrotransposons. The LTRs of wheat retro-

[+]Present address: Department of Medical Parasitology and Infection Biology, Swiss Tropical and Public Health Institute, Basel 4123, Switzerland

[++]Present address: University of Basel, Basel 4001, Switzerland

T. Wicker, C. Stritt[+],[++], A. G. Sotiropoulos, M. Poretti
Department of Plant and Microbial Biology
University of Zurich
Zurich 8008, Switzerland
E-mail: wicker@botinst.uzh.ch
C. Pozniak, S. Walkowiak
Crop Development Centre
University of Saskatchewan
Saskatoon, Saskatchewan SK S7N 5A8, Canada

transposons are typically 0.5–2 kb long and flank an internal domain that encodes canonical proteins such as GAG, reverse transcriptase and integrase. Interestingly, a time-scale can be attached to the TE fossil record of LTR retrotransposons; because of the mechanism of reverse transcription, the two LTRs of a retrotransposon are identical at the time of insertion. Over time, the LTRs accumulate mutations independently, and thus the number of differences between the two LTRs is directly proportional to the time that has passed since the retrotransposon inserted into the genome. Thus, the insertion time (i.e., age) of each individual retrotransposon can be estimated, as long as both LTRs are still present. In plants, a nucleotide substitution rate of 1.3E-8 per site per year[9] is typically applied to obtain insertion age estimates. Because LTR retrotransposons tend to be fragmented by deletions over time estimates of insertion ages for older elements if often not possible due to the lack of one or both LTRs. Thus, that plant genomes contain numerous old TE fragments whose age cannot be estimated anymore.

Many TEs in large genomes are non-autonomous,[1] meaning that they do not encode the full complement of proteins necessary for their replication. However, non-autonomous elements can still be replicated by proteins expressed by autonomous elements located elsewhere in the genome. Some of the most abundant TEs in plants are non-autonomous. For example, miniature inverted-repeat transposable elements (MITEs), which can number in the tens of thousands, contain no coding sequences and are basically reduced to short terminal repeat motifs that are recognized by transposase enzymes. They seem to be cross-activated by only a few autonomous elements usually belonging to the Mariner of Harbinger super-families.[10,11] Non-autonomous elements can parasitize autonomous ones for millions of years. Indeed, the high-copy families *RLG_Sabrina* and *RLG_WHAM* in barley were suggested to be cross-mobilized by *RLG_BAGY2* retrotransposons, despite having very little sequence homology.[12] However, in the absence of experimental data, it is difficult to determine which autonomous TE family cross-mobilizes a particular non-autonomous family.

Bread wheat (*Triticum aestivum*) is one of the world's most important crops, and a high-quality reference sequence of its 16 Gb genome was published a few years ago.[13] At the time of publication, it was one of the largest genomes sequenced to date, due to its high repeat content of at least 85%[8] More recently, chromosome-scale assemblies of nine additional wheat geno-

types became available[14] Additionally, the genomes of 20 barley lines[5,15] and two rye lines[16] close relatives of wheat, recently became available.

Bread wheat is a allohexaploid (genome formula AABBDD) that combines the genomes of the diploid species *T. urartu* (A-genome), *T. tauschii* (D-genome) and a yet unknown donor of the B-genome. Tetraploid wheat formed 400 000–800 000 years ago by bringing together the A and B genomes, while the addition of the D genome to form allohexaploid wheat happened much more recently, ≈10 000 years ago[17] The diploid genome donors, which diverged 3–6 million years ago[17,18] have very similar gene content but differ strongly in their intergenic regions[8] due to the genomic turnover described above. This peculiar evolutionary history and high repeat content make wheat an attractive model system to study TE evolution and dynamics in polyploids, which is now possible due to the availability of high-quality genome assemblies.

In large and repetitive plant genomes, individual TE families often show characteristic chromosomal distributions, resulting in strongly compartmentalized genomes. The distal and telomeric regions are usually enriched in DNA transposons,[19,20] with *CACTA* elements being predominant in wheat and barley[8,12] In contrast, gene-poor regions on chromosome arms and centromeres are populated mostly with LTR retrotransposons of the *Gypsy* and *Copia* superfamilies[8,12,21] It is possible that this distribution is the result of TEs accumulating where they have the least deleterious effects. Only a small number of TEs appear to actively target specific genomic regions. For example, the integrase of *Ty1* elements in yeast has binding affinity to RNA polymerase III, resulting in insertions near genes. Similarly, *Gypsy* retrotransposons of the *RLG_Cereba* clade are found almost exclusively in centromeres of wheat and barley[8,12,22] The unique characteristic of all centromere-specific retrotransposons in grasses is that their integrase contains a chromodomain, which may enable them to target CENH3 histone modifications in active centromeres[23]

Here, we present genome-wide analyses of populations of the most abundant TE families in the allohexaploid wheat genome. We analyze their distributions in wheat sub-genomes and close relatives, as well as their evolution. We found that TE families evolved into distinct sub-populations and sub-families that were active during different time spans, in different sub-genomes and different wheat lineages. Moreover, the data allowed a detailed analysis of the co-evolution of autonomous and non-autonomous retrotransposons.

## 2. Results

### 2.1. The Wheat Genome Contains Tens of Thousands of Full-Length Retrotransposons

For this study, we focused on five previously characterized high-copy LTR retrotransposon families in the wheat genome[8,12] (**Table 1**). All five retrotransposon families were found in all published Triticeae genomes[8,12,16] indicating that they were present already in the Triticeae common ancestor.

Our analyses have the following limitations: we decided to focus only on full-length copies (which contain both LTRs) to allow insertion time estimates. Furthermore, we only used elements with only small InDels, because it is technically extremely chal-

S. Walkowiak
Grain Research Laboratory
Canadian Grain Commission
Winnipeg, Manitoba R3C 3G8, Canada

H. Gundlach
PGSB Plant Genome and Systems Biology
Helmholtz Center Munich
German Research Center for Environmental Health
Neuherberg 85764, Germany

N. Stein
Leibniz Institute of Plant Genetics and Crop Plant Research (IPK)
Seeland 06466, Germany

N. Stein
Center of Integrated Breeding Research (CiBreed)
Department of Crop Sciences
Georg-August-University
Göttingen 37075, Germany

**Table 1.** Number of full-length retrotransposon copies extracted from the *T. aestivum* genome.

| TE family | Superfamily | Full-length | Solo-LTRs | Cons. LTRs[a] | Pol. Sites[b] |
|---|---|---|---|---|---|
| *RLC_Angela* | *Copia* | 19,859 | 7,457 | 18 | 6,005 |
| *RLG_Wilma* | *Gypsy* | 5,331 | 2,148 | 1 | 3,820 |
| *RLG_Sabrina* | *Gypsy* | 15,427 | 7,732 | 7 | 6,084 |
| *RLG_WHAM* | *Gypsy* | 5,676 | 2,410 | 4 | 5,104 |
| *RLG_Cereba* | *Gypsy* | 1,914 | 1,304 | 5 | 1,784 |
| Total | - | - | 47,791 | 21,051 | - |

[a] Number of consensus LTRs used for BlastN searches; [b] number of polymorphic sites found when full-length copies were aligned with a consensus sequence.

lenging to precisely extract TE copies that contain (often multiple) insertions of other TEs. Populations of full-length LTR retrotransposons were extracted by searching for complete LTR sequences that are in the same orientation and at a defined distance. To cover the intra-family diversity, up to 18 different LTR consensus sequences were generated (Table 1). This was done to obtain un-interrupted BLASTn alignments for the LTRs. Candidate full-length retrotransposons were then screened for the presence of respective coding sequences in the sequence between the LTRs. In the wheat reference genome of landrace Chinese Spring[13] we annotated 47 791 full-length LTR retrotransposons, between 1914 and 19 859 per family (Table 1). Additionally, we identified between 1304 and 7732 solo-LTRs per family, which are the product of intra-element recombination[24] The most abundant family, *RLC_Angela*, was also annotated in nine other wheat genomes[14] (Table S1, Supporting Information).

Our method yielded only a few instances where two full-length LTRs were found by chance in the same orientation and approximately at the distance that is expected for the respective retrotransposon family. Length distribution was in a narrow range for all families, except for *RLG_Sabrina*, which comprises sub-populations of derivatives of different sizes (Figure S1, Supporting Information, see below). Furthermore, 80–88% of all identified full-length candidates were flanked by a target site duplication (TSD) with a maximum of two base mismatches. These TSD mismatches were accepted because our previous study showed that TSD production following insertion is highly error-prone (Wicker et al., 2016). *RLG_Cereba* had a lower fraction (64%) of copies with TSDs (Figure S1, Supporting Information). This could be due to the strong enrichment of these elements in the centromeres, which facilitates inter-element recombination and thus joining of LTRs from different retrotransposon copies. This probably also explains the fact that *RLG_Cereba* has the highest proportion of solo-LTRs (Table 1).

The five retrotransposon families were analyzed separately, and each retrotransposon copy was treated as an individual genome. All individual copies were aligned to a consensus sequence of the respective retrotransposon family, analogous to genomes of individuals being aligned to a reference genome. From these alignments, between 1784 and 6005 polymorphic nucleotide positions were called (Table 1). These were converted into variant call (vcf) files and used for principal component analyses (PCAs), which allowed to define sub-populations and/or sub-families (examples in **Figure 1**). In all PCAs done in this study, the first two principal components (PCs) explain 40–60%

of the variation. Thus, we used only PC1 and PC2 to define sub-populations and/or sub-families. Additionally, we estimated the insertion age of each retrotransposon based on LTR divergence. *RLC_Angela* elements were overall the youngest with a mean insertion age of ≈900 000 years (Figure S2, Supporting Information). However, recent improvements in genome assemblies suggested that some complete sequences of very young elements may still be missing from the current assemblies[25] because their identical (or near-identical) LTRs often contain sequence gaps. Thus the actual mean insertion age of *RLC_Angela* elements may be even lower. The *Gypsy* families *RLG_Sabrina*, *RLG_WHAM* and *RLG_Wilma* are about half a million years older and all have similar insertion age distributions (Figure S2, Supporting Information). Less than 0.2% of all retrotransposon copies were older than 4 million years, reflecting findings of previous studies.[5–8]

## 2.2. Many Newly Inserted Retrotransposons May be "Dead on Arrival"

The three retrotransposon families *RLC_Angela*, *RLG_Wilma* and *RLG_Cereba* comprise elements that contain a long open reading frame (ORF) encoding canonical proteins such as GAG, reverse transcriptase (RT) and integrase (INT). In contrast, the entire populations of *RLG_WHAM* and *RLG_Sabrina* elements lack coding sequences (CDS) for RT and INT and therefore have to be considered non-autonomous. *RLC_Angela* is a special case, as only about half of all elements have coding capacity, while the others have highly degenerate CDS and are apparently non-autonomous. These non-autonomous elements will be discussed in detail further below.

The large numbers of autonomous full-length *RLC_Angela*, *RLG_Wilma* and *RLG_Cereba* copies allowed detailed analysis of their CDS and the level to which encoded proteins are intact. Even in populations of autonomous retrotransposons, many copies with disrupted CDS are expected, because TEs accumulate mutations over time. Additionally, reverse transcriptase (which produces the DNA copy of the mRNA) is known to have a high error rate.[26] This may often result in non-functional copies that have defective CDS containing frame shifts or in-frame stop codons. We mapped consensus sequences for the predicted polyproteins onto the individual copies in order to obtain individual polyprotein sequences. Here, we used the number of in-frame stop codons as measure of CDS degeneration, which was plotted

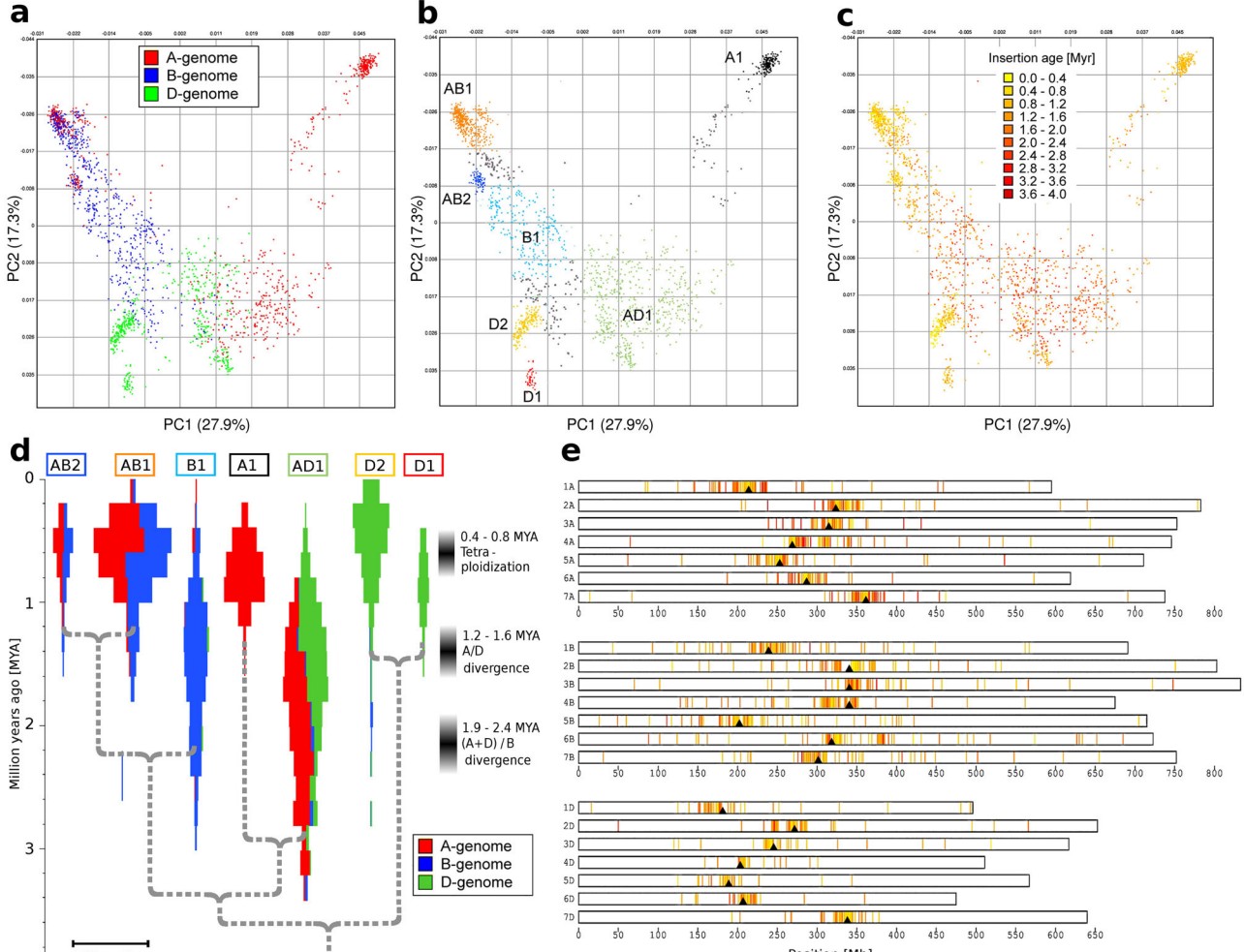

**Figure 1.** Population analysis of *RLG_Cereba* retrotransposons. a) Principal component analysis (PCA) of the sequences of 1914 full-length *RLG_Cereba* element. Each dot corresponds to one retrotransposon. The colors indicate the wheat sub-genome in which each copy resides. The percentage of variation explained by each PC is given in parentheses. b) Classification of *RLG_Cereba* elements into sub-families (shown in different colors) based on PCA. retrotransposon copies that were not assigned to a sub-family are in gray. Note that sub-families A1, B1, D1, and D2 are specific to a single sub-genome while AB1, AB2, and AD1 are found in at least two sub-genomes. c) PCA with TE copies colored according to their insertion age. d) Model for the evolution of *RLG_Cereba* sub-families. Population sizes of *RLG_Cereba* sub-families over time are shown by violin plots. The proportion of elements in the three sub-genomes is indicated with colors red blue and green. Superimposed is a phylogenetic tree of *RLG_Cereba* sub-families (gray dotted line). Divergence times were estimated from synonymous substitutions in CDS (see Figure S4, Supporting Information). e) Localization and insertion ages of *RLG_Cereba* elements along wheat chromosomes. The *x*-axis indicates the position in Mb, the *y*-axis indicates the chromosome. Previously estimated positions of centromeres are indicated with black triangles. Each vertical bar represents one *RLG_Cereba* copy. Insertion ages of the elements are color-coded as in (c).

against the insertion age estimates. Linear regression was then used to estimate the expected number of in-frame stop codons in newly inserted elements (Figure S3, Supporting Information). According to this analysis, newly inserted *RLC_Angela* and *RLG_Cereba* elements contain on average ≈0.3 in-frame stop codons (Figure S3, Supporting Information). In other words, about 30–40% of all newly inserted *RLC_Angela* and *RLG_Cereba* retrotransposons may be dead on arrival copies. For *RLG_Wilma*, a majority may be dead on arrival as they contain an average of 1.4 in-frame stop codons. Interestingly, the slope of the regression line for *RLG_Cereba* is steeper than that for *RLC_Angela* (Figure S3, Supporting Information), and *RLG_Wilma* elements generally contain higher numbers of stop codons.

## 2.3. The History of Wheat Polyploidization is Reflected in Centromeric Retrotransposons

In total, we identified 1914 full-length *RLG_Cereba* elements. Most of them are located in centromeric regions (Figure 1), as was expected[8] Interestingly, the youngest elements tend to occupy the centers of the predicted centromeres, suggesting that they indeed actively target the CENH3 modified histones in functional centromeres[23] resulting in older elements being "pushed" towards the outer limits of the centromere. Nevertheless, there are dozens of full-length *RLG_Cereba* elements spread along chromosome arms, indicating that "off target" insertions occur frequently.

PCA shows that the *RLG_Cereba* family comprises several sub-families, most of them specific to one or two wheat sub-genomes (Figure 1a). For subsequent analyses, we defined seven *RLG_Cereba* sub-families based on the PCA (Figure 1b). Subfamilies were defined by hand in the PCA. Thus, in some cases, the borders are somewhat arbitrary, when boundaries between the sub-families are blurred (Figure 1b). In the PCA, the oldest retrotransposons are in the center, while younger sub-families are at the periphery, reflecting their divergence over time (Figure 1c). PCA and insertion ages also indicate that all present day *RLG_Cereba* sub-families evolved from a retrotransposon population that was present already in the common ancestor of the A, B and D sub-genomes (Figure 1a–c). This means that the oldest elements spread already in the common ancestor of the A, B and D sub-genomes, while the younger ones differentiated only after the A/B/D divergence (Figure 1a,c).

Based on nucleotide substitutions in synonymous sites of the CDS, we estimate that the seven sub-families diverged at various times between 1.3 and 3.7 million years ago (Figure 1d, Figure S4, Supporting Information), and are therefore spread differently across the three sub-genomes. Sub-family AD1 was mainly active 1–3 million years ago and is found in all three wheat sub-genomes (although at low abundance in the B genome), because it was already active before the A, B and D sub-genome progenitors diverged[18] (Figure 1d). In contrast, subfamily B1 only became active after the B genome diverged from the other two, and is therefore found exclusively in the B genome (Figure 1d). Sub-families AB1 and AB2 subsequently evolved from B1 and initially spread across the diploid B genome. When A and B sub-genomes joined to form a tetraploid 400 000–600 000 years ago,[18] they also invaded the A genome (Figure 1d). Finally, the recently emerged sub-families D1 and D2 are restricted to the D genome, and had little opportunity to spread to the other two sub-genomes, because the D genome was added only about 10 000 years ago to form allohexaploid wheat.[17,27]

### 2.4. The *RLC_Angela* Family Provides Insight into Ancient Chromosome Rearrangements

The *RLC_Angela* family is the most abundant TE family in wheat[8] contributing over 10% of the total genome. Similarly, its homolog *RLC_BARE1* in barley contributes about 10% to that genome,[12] indicating that this family has been prolific at least since wheat and barley diverged about 8–10 million years ago[18] PCA shows that the *RLC_Angela* family is very heterogeneous, with the 19 859 identified full-length copies forming five distinct sub-populations (α through ε, **Figure 2**a). Here, we distinguish "sub-populations" from the above described "sub-families" in that they are completely separate from each other, while sub-families form more of a continuum, reflecting how they evolved and diversified from one another. Molecular dating of synonymous sites indicates that the sub-populations were likely present already in the Triticeae ancestor, and started diverging over 20 million years ago (Figure 2b, Figure S5, Supporting Information). Sub-population β was the most recently active (300 000–400 000 years ago), while δ is the oldest with an activity peak 0.9–1.0 million years ago (Figure 2b). The sub-populations β and δ are present in all three sub-genomes, while sub-population α is

mostly found in the D genome (example in Figure 2c, Figure S6, Supporting Information), indicating that it had an activity burst after the D genome lineage branched off about 1.2–1.6 million years ago.

We further divided the five *RLC_Angela* sub-populations into 1–10 sub-families (examples in Figure 2d,e, Figure S7, Supporting Information). Copies from the same sub-population are roughly 89–98% identical, while those from different sub-populations share only 80–90% sequence identity. Of these, sub-population δ is the most interesting: on one hand, it contains autonomous and non-autonomous elements (see below). On the other hand, its ten sub-families are highly sub-genome specific because they were already active in the diploid progenitors of wheat (Figure 2b,f,h) and went largely silent before the formation of tetraploid wheat 400 000–800 000 years ago[17] In contrast, retrotransposons of sub-population β were active more recently, peaking 300 000–600 000 years ago (Figure 2f), in the time after the A and B sub-genomes had merged (Figure 2g).

Because of their different activity peaks, sub-family profiles can be used to characterize chromosomal translocations and introgressions. Indeed, retrotransposons of the δ1a and δ1n sub-families are highly enriched at the ends of chromosomes 2A and 6A (I segments, Figure 2h), indicating that these chromosomal segments were introgressed from a different genetic background, possibly from a wheat relative. Subfamily δ1a and δ1n retrotransposons are also found elsewhere in the wheat genome, but the putative introgression segments contain many more younger copies, indicating that they were active in more recent times in the donor species (Figure S8, Supporting Information). Similarly, a long-known translocation of ≈100 Mb from chromosome 7B to 4A[28] is clearly marked by the presence of subfamily δ5a and δ5n elements which are otherwise found mostly in the B genome (segment II, Figure 2h). This analysis also identified an additional small introgression of a B genome segment into chromosome 2A (segment III, Figure 2H, Figure S8, Supporting Information). Interestingly, these patterns are not visible in the distribution of the more recently active sub-population β retrotransposons (e.g., sub-families β6 or β8), which are evenly distributed across A and B genomes (Figure 2g). Thus, one can date the 7B/4A translocation to a period after sub-population D elements went silent and before sub-population sub-families β6 or β8 had their major activity burst 300 000–400 000 years ago. This leaves a narrow window, indicating that the translocation happened soon after tetraploidization 400 000–800 000 years ago[17]

### 2.5. *RLC_Angela* Populations Can be Used to Characterize Chromosomal Introgressions in the Wheat Germplasm

In a recent study, we analyzed the genomes of ten wheat lines and found that individual lines often contain chromosomal segments that contain many unique *RLC_Angela* insertions. In other words, these segments have a retrotransposon "insertion history" that differs from that of the other sequenced wheat lines. These chromosomal segments were interpreted as distinct haplotype segments and were, in some cases, demonstrated to be introgressions from a tertiary gene pool of wheat.[14] For this current study, we wanted to further characterize these introgressions to obtain clues as to their origin. We extracted *RLC_Angela*

**2100022 (5 of 17)**

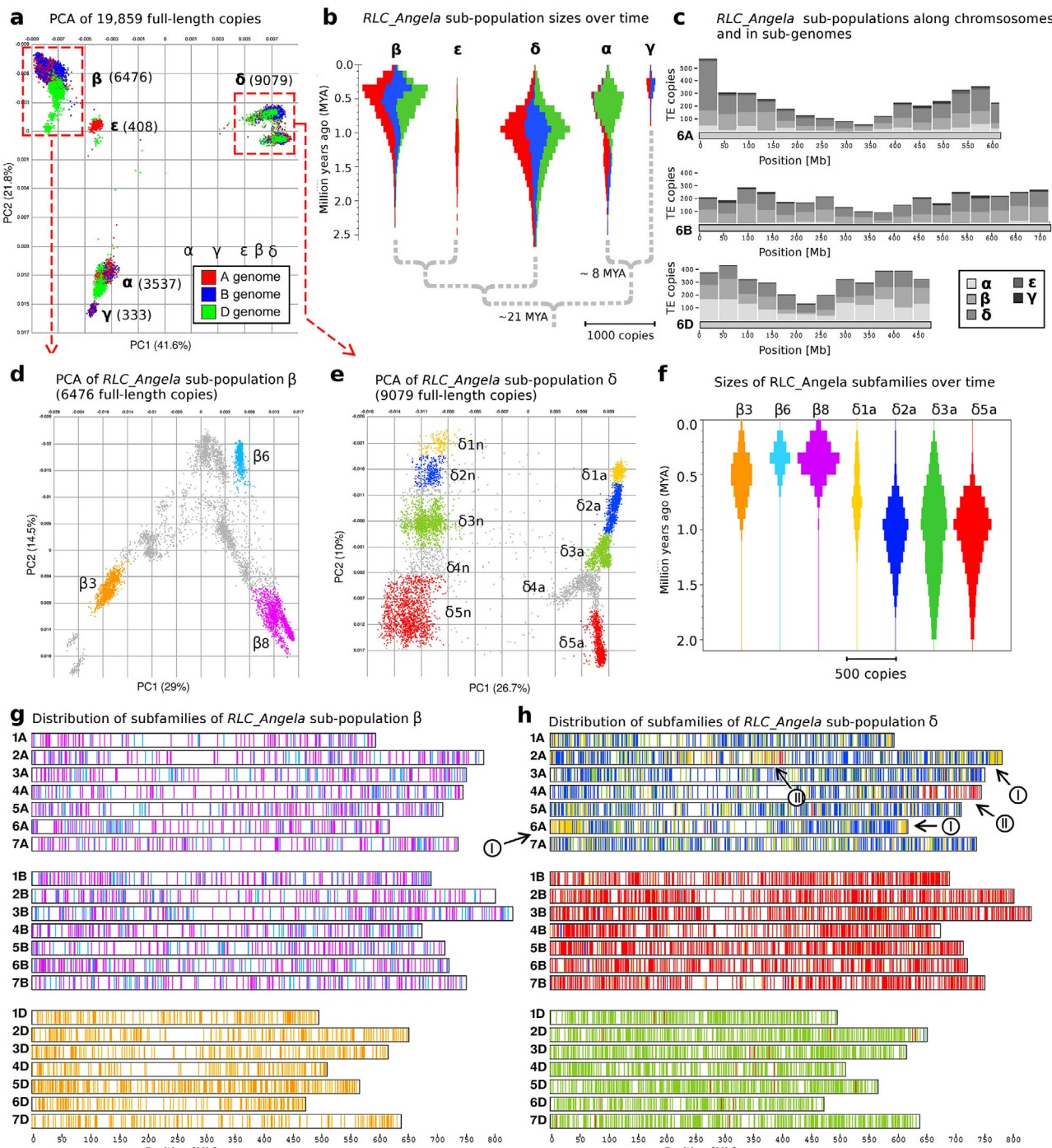

**Figure 2.** Analysis of *RLC_Angela* retrotransposons. a) PCA of 19 859 full-length retrotransposons, each colored according to the wheat sub-genome in which it was found. In all PCAs, the percentage of variation explained by each PC is given in parentheses. b) Violin plots of sizes of *RLC_Angela* sub-populations over time. The proportions in each sub-genome are colored as in (a). Phylogenetic relationships between sub-populations are indicated with dashed lines. Divergence times of lineages was inferred from synonymous CDS sites (see Figure S5, Supporting Information). c) Distribution of *RLC_Angela* sub-populations along wheat chromosomes in bins of 40 Mb. Group 6 chromosomes are shown as representative examples. d) and e) Examples for definition of *RLC_Angela* sub-families based on PCAs of sub-populations $\beta$ and $\delta$, respectively. f) Violin plots showing the sizes of sub-families over time. Colors and sub-family names correspond to those in (d) and (e). g) Distribution of sub-families of sub-population $\beta$ along wheat chromosomes. Individual TEs are indicated as vertical bars with colors corresponding to those in (d) and (e). h) Distribution of sub-families of sub-population $\delta$ along wheat chromosomes. Colors correspond to those in (b) and (c). Sub-families $\delta$1a and $\delta$5a are highly enriched on a few chromosomal segments (I, II, and III), identifying a translocation and possible introgressions. Note that these patterns are not discernible in the distribution of the younger sub-population $\beta$, indicating the introgressions and translocation occurred before sub-population $\beta$ retrotransposons invaded the wheat genome.

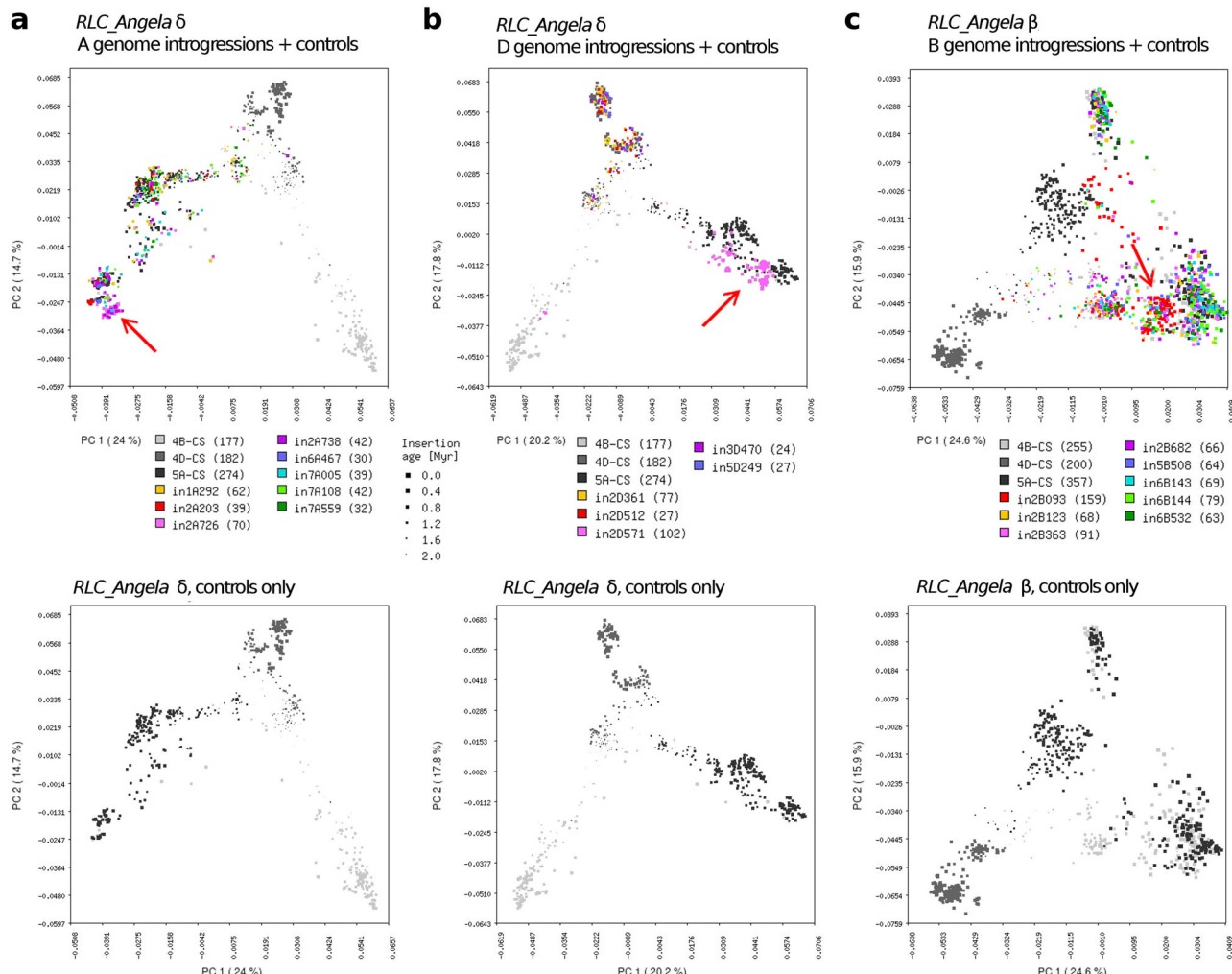

**Figure 3.** Characterization of chromosomal introgressions in wheat using PCAs of *RLC_Angela* retrotransposon using PCA. The percentage of variation explained by each PC is given in parentheses. In each top panel, retrotransposons from identified introgressions are shown as colored dots with younger elements being indicated as larger dots. In shades of gray are retrotransposons from the A, B, and D sub-genomes from introgression-free segments of the wheat genome backbone, serving as controls. The bottom panels show only the A, B, and D genome controls to make it easier to see retrotransposons that were introgressed from different genetic backgrounds in the top panels. Introgressions names have the prefix "in" followed by the chromosome name and the start point of the introgression (in Mb). a) *RLC_Angela* sub-population δ elements in A genome introgressions. Most introgressions come from backgrounds that are not distinguishable from the A genome. However, Introgressions in2A726 and in2A730 form a unique cluster (red arrow) which is near other A-genome copies, but is still distinct from them. This indicates that in2A726 and in2A730 originate from a close A-genome relative. b) *RLC_Angela* sub-population δ elements in D genome introgressions. Note that in2D571 comes from a donor that is similar to the A genome, while all other introgressions come from D genome hayplotypes. c) *RLC_Angela* sub-population β elements in B genome introgressions. Note that retrotransposons from in2B093 form a unique cluster in an intermediate position between A and B genome retrotransposons (red arrow), suggesting that donor of the introgression is equally distantly related from the A and B genomes.

copies from all predicted introgressions that contained at least 40 elements either of the *RLC_Angela* β or δ sub-populations. The A and B genomes contain each eight such introgressions, while the D genome contains only five (Table S2, Supporting Information). As control, we used *RLC_Angela* retrotransposons that come from chromosomes 5A, 4B and 4D since these contain only a few short introgressions in the Chinese Spring reference genome (retrotransposons coming from these introgressions were removed).

Chromosomal introgressions in the A and D genomes showed distinct signatures in *RLC_Angela* elements of the sub-population δ (**Figure 3**). The retrotransposons of the 2A726 and 2A738 introgressions (in wheat lines Norin 61 and CDC Land-mark, respectively) form a distinct group that is different from the control elements from the A genomes. This indicates that these two introgressions come from a relative of the wheat A genome that is distant enough so that a distinct lineage of *RLC_Angela* has evolved in it (Figure 3a). Retrotransposons from all the other A genome introgressions cluster in the PCA with control A genome elements, indicating that they originated from very closely re-lated A-genome relatives (Figure 3a). Similarly, four of the five D genome introgressions are clearly rooted in other D genome

lineages, as their retrotransposons closely cluster with the D genome reference elements (Figure 3b). The most interesting, however, is introgression 2D571 which relates to a donor that was close to, but still distinct from the A genome (Figure 3b, Figure S9, Supporting Information). These signatures in A and D genome introgressions are only visible in *RLC_Angela* elements of the more ancient sub-population $\delta$, while sub-population $\beta$ elements cluster with their corresponding control retrotransposons (Figure S9, Supporting Information).

Introgressions into the B genome show distinct PCA signatures only when the *RLC_Angela* sub-population $\beta$ is used (Figure 3c, Figure S9, Supporting Information). Here, the retrotransposons found in the introgression 2B093 in wheat line LongReach Lancer form a separate cluster that is close to some B-genome specific elements (Figure 3c). This particular introgression was shown to come from the tetraploid *Triticum timopheevii* and has important implications for breeding and disease resistance[14] *T. timopheevii* is a tetraploid wheat that formed independently and combined species containing the sub-genomes A and G. Indeed, the G genome of *T. timopheevii* was placed closest to the B genome in phylogenetic analyses,[29,30] a result that is accurately reflected in the PCA of *RLC_Angela* elements. *RLC_Angela* elements from all the other B genome introgressions cluster with other B genome elements, indicating that they originated from other B genome relatives (Figure 3c).

## 2.6. The *RLC_Angela* Family Contains Several Co-Evolving Groups of Autonomous and Non-Autonomous TEs

In the *RLC_Angela* family, the putative non-autonomous elements are similar in size to the autonomous ones and over 85% identical at the DNA level. Here, we refer to autonomous sub-populations when they, in principle, encode elements with intact CDS, even if they accumulated mutations over time. In contrast, the non-autonomous elements have a highly degenerated CDS with numerous frame-shifts and in-frame stop codons and clearly do not encode functional proteins (**Figure 4**a). The *RLC_Angela* sub-population $\delta$ contains both autonomous and non-autonomous elements, and these two groups are clearly separated by the first principal component in the PCA (Figure 4b), while the second principal component (PCA $y$-axis) separates the five autonomous ($\delta$1a through $\delta$5a) and five non-autonomous sub-families ($\delta$1n through $\delta$5n, Figure 4b). The pairing of non-autonomous and autonomous sub-families in the second principal component of the PCA suggested that they are functionally linked. Indeed, these putative pairs have virtually identical insertion age distributions and sub-genome specificities (Figure 4c), indicating that they were active at the same time and in the same sub-genome(s).

Insertion age distributions show that the *RLC_Angela* sub-population $\delta$ was active earlier (Figure 2b). Additionally, it is the only sub-population that contains both autonomous and non-autonomous elements (Figure 4b). In contrast, sub-population $\beta$ consists exclusively of non-autonomous elements, while sub-populations A and F comprise only autonomous ones. A phylogenetic tree constructed from internal domains of *RLC_Angela* elements indicates that all non-autonomous sub-families of sub-populations $\beta$, $\delta$, and $\epsilon$ are monophyletic and evolved from au-

tonomous elements of sub-population IV (Figure 4d). In contrast, the autonomous sub-populations $\alpha$ and $\delta$ are more basal in the tree. Thus, it appears that they split into non-autonomous and autonomous elements that occurred originally in sub-population $\delta$, from which later the purely non-autonomous sub-populations $\beta$ and $\epsilon$ evolved.

Interestingly, the phylogenetic tree of LTR consensus sequences (Figure 4e) has a very different topology than that derived from internal domains (Figure 4d). To be active at the same time, non-autonomous elements have to be expressed together as their autonomous partners. Therefore, the LTRs, which contain regulatory sequences of non-autonomous elements, have to co-evolve with those of autonomous ones. Indeed, a phylogenetic tree constructed with LTR consensus sequences from *RLC_Angela* sub-population $\delta$ elements shows that LTRs of the predicted autonomous/non-autonomous pairs strictly cluster together (Figure 4e). In contrast, the more "modern" non-autonomous elements from sub-populations $\beta$ and $\epsilon$ do not have such clear autonomous partners. In fact, in the LTR tree sub-population $\beta$ and $\epsilon$ elements cluster together with autonomous $\alpha$ and $\gamma$ elements (Figure 4e), suggesting that $\alpha$ and $\gamma$ are the autonomous partners.

From the combined data we conclude: i) two ancestral lineages of autonomous *RLC_Angela* retrotransposons diverged at least 20 million years ago, one leading to sub-populations $\alpha$ and $\gamma$, and the other to sub-populations $\beta$, $\delta$, and $\epsilon$. ii) Non-autonomous elements evolved in sub-population $\delta$. From these, non-autonomous sub-populations $\beta$ and $\epsilon$ later evolved later. iii) About 500 000 years ago, the autonomous retrotransposons of sub-population $\delta$ went largely silent, halting the spread of their non-autonomous partners. Interestingly, sub-populations $\beta$ and $\epsilon$ (which presumably were originally dependent on $\delta$ elements) somehow transitioned to being parasites of sub-population $\alpha$ and $\gamma$ elements, as the phylogenetic tree of LTRs suggests (Figure 4e).

## 2.7. *RLG_Sabrina* and *RLG_WHAM* are Co-Evolving High-Copy *Gypsy* Retrotransposons

*RLG_Sabrina* retrotransposons are among the most abundant elements in wheat and barley[8,12] contributing ≈9% to both of these genomes. As mentioned above, *RLG_Sabrina* elements are non-autonomous, since they do not encode a full polyprotein. As in barley[12] the *RLG_Sabrina* family in wheat is comprised of two main groups, one with longer elements (8.1–8.3 kb) and one with shorter ones (6.6–7.6 kb, **Figure 5**). The long elements (*RLG_Sabrina_SA*, Figure 5b) identified in this study encode a GAG protein and a protease domain immediately downstream of GAG (here referred to as GAG-Pro, Figure 5b). The presence of this intact GAG-Pro ORF suggests that *RLG_Sabrina_SA* contributes some functional proteins during its replication cycle, and thus may be considered "semi-autonomous" (hence the "SA" extension in its name). The shorter *RLG_Sabrina* elements can be grouped into at least seven variants, all of which are present in several hundred copies in the wheat genome (Figure 5b). The difference to the semi-autonomous *RLG_Sabrina_SA* elements is that the region of the *GAG-Pro* gene is replaced by completely different sequences (Figure 5b) which encode short proteins. Four of these derivatives encode short proteins

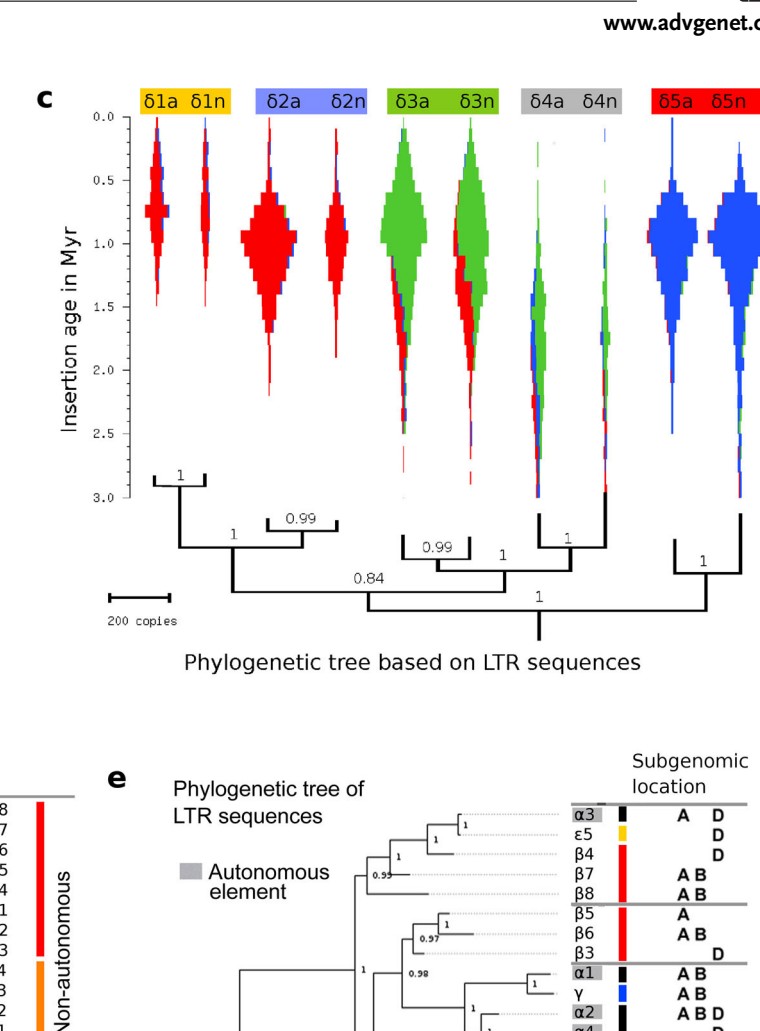

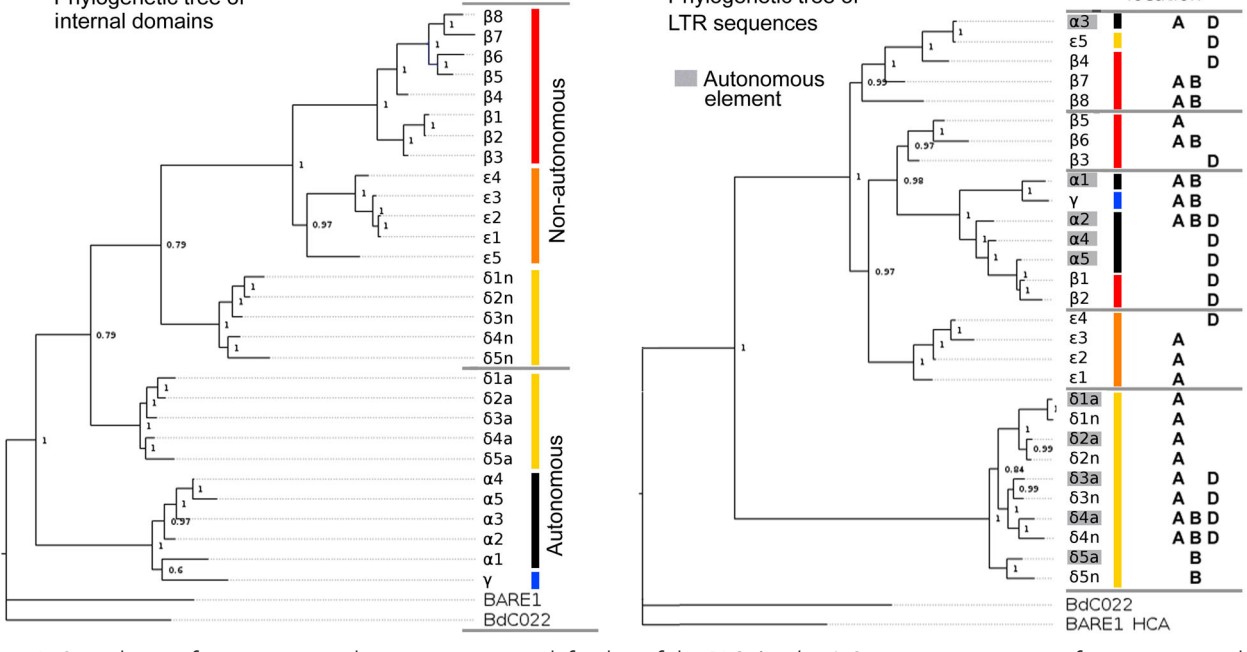

**Figure 4.** Co-evolution of autonomous and non-autonomous sub-families of the *RLC_Angela*. a) Sequence organization of autonomous and non-autonomous *RLC_Angela* elements. b) Definition of *RLC_Angela* sub-families based on PCAs of sub-population δ. The appendix "a" stands for autonomous, while "n" stands for non-autonomous. c) Violin plots of activity of sub-families *RLC_Angela* sub-population δ across time and in different wheat sub-genomes. sub-family names and underlying colors correspond to those in Figure 2e. Note that putative autonomous/non-autonomous pairs have virtually identical distributions. d) Phylogenetic tree of internal domain sequences of all identified *RLC_Angela* sub-families. Sub-families from individual sub-populations (indicated by name prefixes and vertical colored bars) cluster strongly. Sequences from barley and *Brachypodium* homologs were used as outgroups. e) Phylogenetic tree of LTRs of the same sub-families as in (d). Clustering of LTRs from autonomous and non-autonomous elements can indicate which autonomous sub-families cross-mobilize which non-autonomous sub-families. Note that in sub-population δ, pairs of autonomous and non-autonomous sequences that were identified by PCA (Figure S2, Supporting Information) cluster consistently.

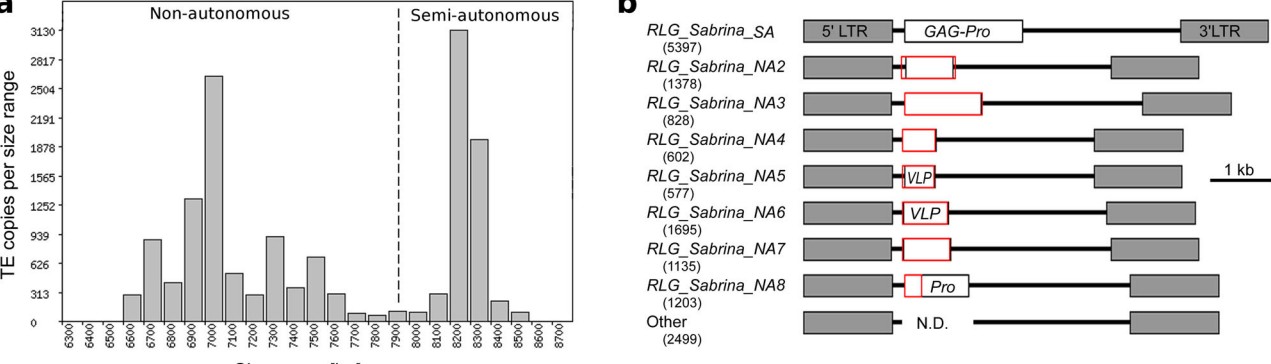

**Figure 5.** Analysis of the *Gypsy* family *RLG_Sabrina*. a) The 15 427 full-length *RLG_Sabrina* elements were split into semi-autonomous (SA) and non-autonomous (NA) groups based on their size distribution (dashed line). b) Sequence organization of *RLG_Sabrina* variants. *RLG_Sabrina_SA* is the variant with the most coding capacity as it contains a CDS for GAG and proteinase. The shorter variants (*RLG_Sabrina_NA2* through *NA8*) contain unique segments encoding other short proteins instead of the GAG-Pro gene (red boxes). *RLG_Sabrina_NA9* encodes a partial proteinase protein. The identified number of full-length copies is given in parentheses for each variant. Pro: Poteinase, VLP: Virus-like particle, homology to proteins that are evolved in viral capsid formation.

without homology to any other wheat proteins. Interestingly, two (*RLG_Sabrina_NA5* and *RLG_Sabrina_NA6*) encode proteins that show strong similarities at the 3D structural level to proteins that are involved in capsid formation in viruses, suggesting that these sequences were acquired from viruses. Due to the extreme diversity of sequences, we chose to focus only on the 5397 semi-autonomous *RLG_Sabrina* elements for further analysis. We categorized these into eight sub-families based on PCA (Figure 5a).

The *RLG_WHAM* family has similar copy numbers as *RLG_Sabrina* but is less diverse. Similar to *RLG_Sabrina_SA*, the *RLG_WHAM* retrotransposons only encode a protein with GAG and proteinase domains, while RT and INT are absent. Using PCA, we defined eight sub-families, of which the younger ones again are largely sub-genome specific (**Figure 6**b). Interestingly, the PCAs and insertion age distributions of *RLG_WHAM* and *RLG_Sabrina* are very similar (Figure 6), suggesting co-evolution of these two families (see below).

### 2.8. *RLG_Wilma* is the Likely Autonomous Partner of *RLG_Sabrina* and *RLG_WHAM*

The large sequence dataset from TE populations allowed us to scrutinize the previously described hypothesis that *RLG_Sabrina* and *RLG_WHAM* are cross-mobilized by *RLG_BAGY2*-like elements[12]. The wheat homolog of *RLG_BAGY2*, for historical reasons called *RLG_Wilma*, is a high-copy family for which we annotated 5331 full-length copies. Sequence homology between *RLG_Wilma*, *RLG_Sabrina*, and *RLG_WHAM* families is minimal, indicating that they diverged a long time ago. Indeed, we identified a homolog of *RLG_WHAM* in *Brachypodium*, which diverged from wheat 35–45 million years ago[17,18] thereby supporting that these elements were present early on in grasses. If *RLG_Wilma* is indeed the autonomous partner of *RLG_Sabrina* and *RLG_WHAM*, the following five criteria would have to be met: i) PCAs of autonomous and non-autonomous families should look similar with respect to overall distribution and num-

ber of sub-groups/sub-populations (examples in Figure 6), ii) insertion age distribution should show that autonomous and non-autonomous elements were active at the same time, iii) the primer binding site (PBS) downstream of the LTR should be similar so that the same tRNA primer can be used by the reverse transcriptase, iv) LTR termini should be conserved because they serve as binding sites for integrase,[31] and v) because non-autonomous and autonomous elements must be expressed at the same time, promoter sequences should show at least some sequence conservation.

Although there is hardly any sequence homology between *RLG_Wilma*, *RLG_Sabrina*, and *RLG_WHAM* (**Figure 7**), they fulfill all five criteria. Indeed, PCAs of the three families show very similar sub-family structures and distributions across wheat sub-genomes (Figure 6a-c). In fact, for every distinct sub-family in the PCA of *RLG_Wilma* elements, there is a corresponding sub-family in the PCAs for *RLG_WHAM* and *RLG_Sabrina* (Figure 6). Furthermore, the corresponding sub-families of *RLG_Wilma*, *RLG_Sabrina* and *RLG_WHAM* elements were active during the same periods and in the same sub-genomes (Figure 6). Additionally, the PBS are identical in all three families, while sequence conservation in the neighboring regions is very low (Figure 7d). Interestingly, the predicted PBS is identical to the 3'end of the tRNA[Asp] from grasses, indicating that this tRNA serves as primer for the reverse transcription. Analysis of LTRs showed that 5' termini are highly conserved, with the first nine nucleotides being identical in all three families (Figure 7c), while at the 3' end, only the canonical terminal CA motif is conserved. Finally, the LTRs contain a conserved GC-rich motif that is followed by a potential TATA box in their 5' region (Figure 7e).

### 2.9. Evidence for Current and Recent Retrotransposon Activity is Sparse

We searched the datasets available to us for evidence of recent or ongoing successful replication of the five retrotransposon families. We performed a phylogenetic analysis and identified

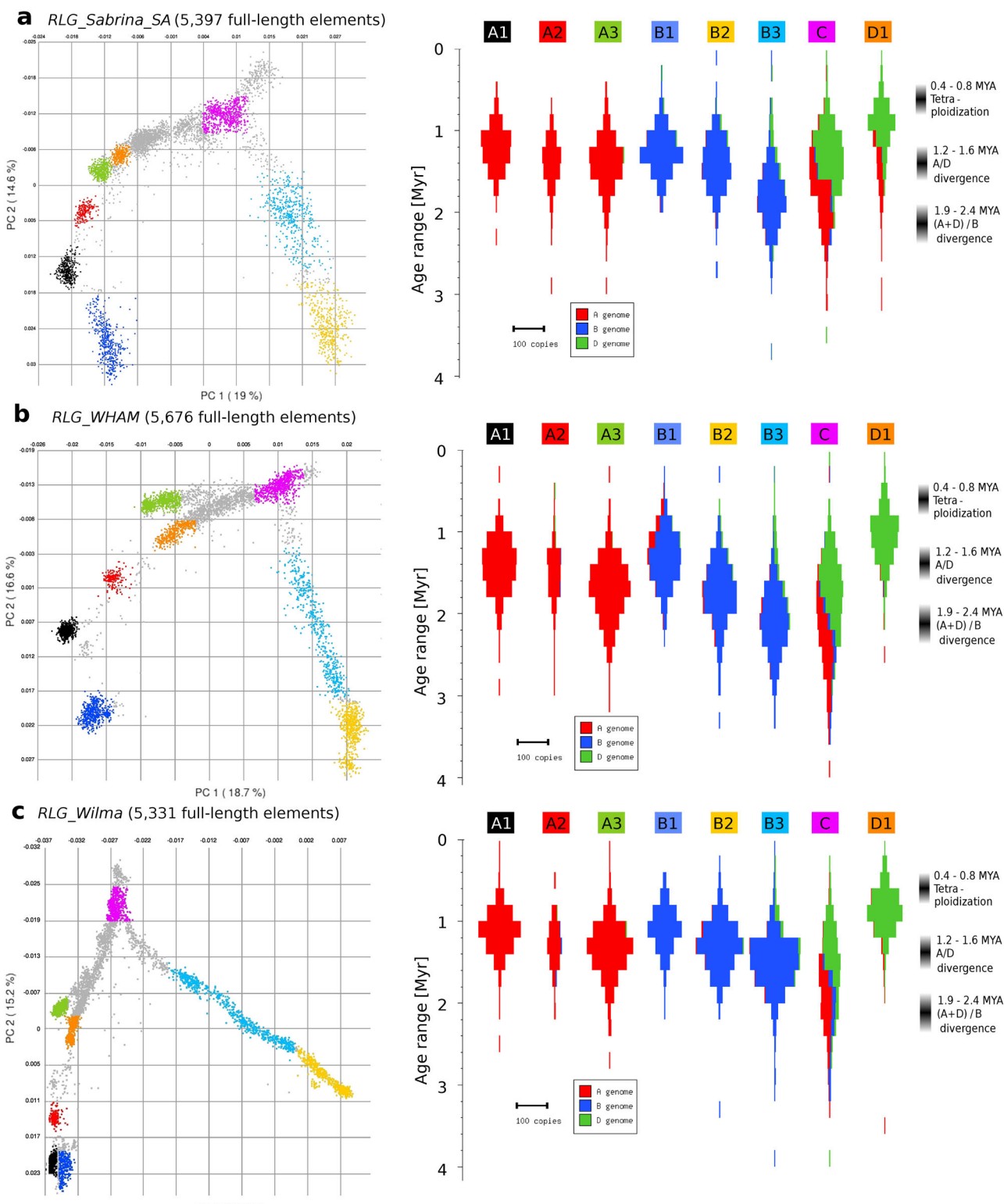

**Figure 6.** Comparative analysis of a) *RLG_Sabrina*, b) *RLG_WHAM*, and c) *RLG_Wilma* elements to support the hypothesis that *RLG_Wilma* is the autonomous partner that mobilized the other two. The left panels show PCAs of the three families, where individual TE copies are colored according to sub-families they were classified in. Note that the PCAs are overall very similar between the three TE families, suggesting that they co-evolved. The panels at the right show violin plots of insertion ages and sub-genome distributions. Time estimates for divergence of sub-genomes and tetraploidization are shown at the right. Note that the sub-families that correspond in the three PCAs also have very similar insertion age and sub-genome distributions.

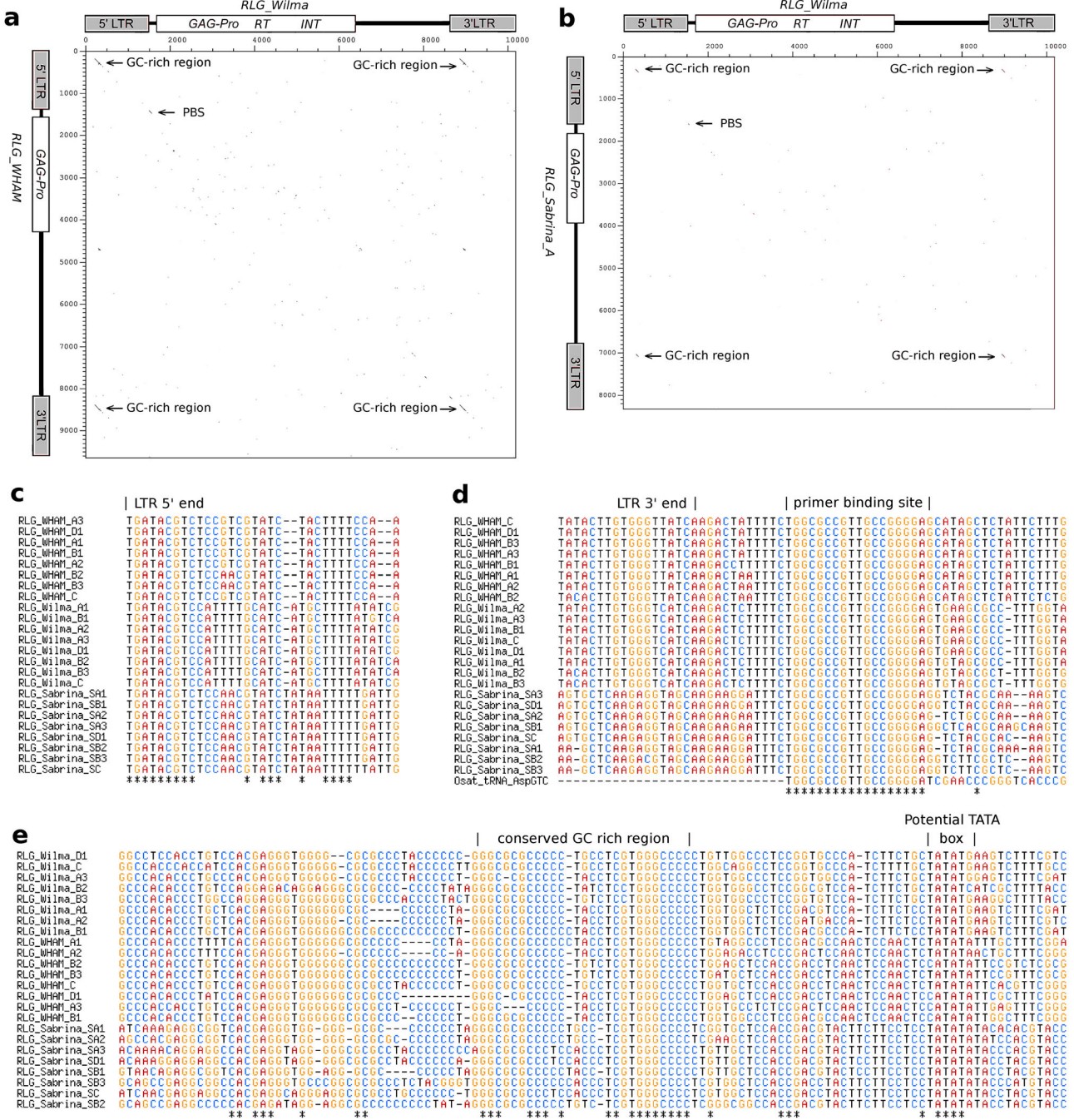

**Figure 7.** Identification of conserved motifs in sequences of *RLG_Wilma*, *RLG_WHAM*, and *RLG_Sabrina*. a,b) Dot plot comparisons of the autonomous *RLG_Wilma* retrotransposon (horizontal) with its putative non-autonomous partners a) *RLG_WHAM* and b) *RLG_Sabrina*. Note that there is almost no detectable sequence conservation except from a short GC-rich motif in the LTR and the primer binding site (PBS) just downstream of the LTR. c) Sequence alignment of the 5' end of LTRs. Integrase specifically recognizes the termini of LTRs. These are identical in all three retrotransposon families and perfectly conserved in all sub-families. d) Sequence alignment the 3'end of the LTR and the primer binding site. Note that sequences between families vary strongly, except for the PBS, which is perfectly conserved in all families and sub-families, indicating that replication of all three families is primed with the same tRNA. It is likely that tRNA$^{Asp}$ serves as primer, as the PBS is identical to the 3'end of tRNA$^{Asp}$ from rice (bottom sequence). e) Sequence alignment of the conserved GC-rich motif with a potential downstream TATA box. Except for the termini, this region is the only one that has significant sequence homology between the three retrotransposon families.

only two clades containing six retrotransposons in the D-genome which are descendants of A/B genome elements (Figure S12, Supporting Information). This indicates A and B-genome retrotransposons were active to some degree after the formation of hexaploid wheat and some inserted into the D-genome. Additionally, we searched the ten wheat genomes for retrotransposon copies which have identical LTRs (i.e., they inserted so recently that they accumulated no mutations) and which are present exclusively in one wheat line. We identified only four copies for which we could confirm the specific recent insertion by identifying the orthologous locus in other wheat genomes which do not contain the retrotransposon copy (Table S3, Supporting Information). All four belong to the recently active *RLC_Angela* sub-population *β*.

Finally, we searched our recently published wheat transcriptome data[32] for evidence of transcription. Interestingly, we found high numbers of transcriptome reads mapping to the five retrotransposon families in all four conditions and all replicates, with *RLG_WHAM* having the highest and *RLG_Wilma* having the lowest numbers (Figure S12). This finding is curious, considering that we found only very few recently inserted *RLG_WHAM* copies. One possible explanation is that the autonomous partner, *RLG_Wilma*, has much lower expression levels (Figure S12b, Supporting Information). Additionally, the *RLG_Wilma* population contains only few copies with intact coding regions (Figure S12c, Supporting Information). Indeed, none of the copies younger than 100 000 years have an intact ORF. In contrast, *RLC_Angela* and *RLG_Cereba* for which we found many young copies contain also much higher numbers of intact ORFs (Figure S12c, Supporting Information).

As mentioned above, recent improvements in assembly technology indicated that very recently inserted retrotransposon copies are more likely to contain sequence gaps[25] and may therefore be less well represented in our survey. Even if this is the case, our data indicate that the five high-copy retrotransposon families studied here highly successfully populated the wheat genome in the past 2–3 million years, but showed only very low levels of activity in past few hundred thousand years.

# 3. Discussion

Analysis of retrotransposon populations in wheat provided detailed insight into the evolutionary dynamics of retrotransposons as well as into the evolution of the wheat genome. Wheat is an excellent system for such studies because of its high repeat content and its allohexaploidy. Applying a combination of population genetics methods, phylogenetic analyses and molecular dating provided a simple and yet precise way to distinguish TE sub-populations and sub-families. Although such detailed classification of repetitive sequences may seem a bit excessive at first glance, it is the TE sub-family level that revealed the most interesting biological insights. On one hand, we could show that TE sub-families can be used as "evolutionary markers" for the identification of chromosomal introgressions and rearrangements because individual sub-families may have different levels of activity in different wheat lineages. On the other hand, analysis of retrotransposon sub-families allowed an in-depth analysis of the co-evolution of autonomous and non-autonomous elements. Both

types of analyses are, to our knowledge, novel in plants and are therefore discussed further hereafter.

## 3.1. Population Analysis of TEs Sheds Light on the History of Chromosomal Introgressions and Rearrangements

In a recent study, we were able to identify hundreds of candidate introgressions in wheat based on their specific composition of *RLC_Angela* retrotransposons[14] Here, we further studied some of the large introgressions and aimed at identifying possible donor species. Comparison of TE populations from introgressed segments with those from all three wheat sub-genomes allowed identification of overall genomic similarities in the absence of sequence information of the actual donor species. We showed that most introgressed chromosomal segments come from closely related wheat lines, which, nevertheless, they were distinct enough that they could be identified due to their high content of unique TE insertions. Most importantly, we were also able to identify several introgressions that came from very different genetic backgrounds. The most remarkable one is 2D571, where an ≈48 Mb segment from an A genome relative was introgressed into the long arm of chromosome 2D. This is highly unusual, since recombination between sub-genomes is extremely rare in wheat, likely due to activity of genes that ensure proper pairing of chromosomes during meiosis such as Ph1 and Ph2.[33,34] Additionally, our data can be used to estimate divergence times of wheat sub-genomes and sub-species: our previous study showed that the nearly chromosome-length 2B093 introgression in the wheat cultivar LongReach Lancer came from the allotetraploid wheat *T. timopheevii*[14] whose G sub-genome is closely related to the B sub-genome of wheat[29,30]. Because retrotransposons from the 2B093 introgression only show a distinct PCA signature in the younger *RLC_Angela β* sub-population, but not in the older *δ* sub-population, our results suggest that the B and G sub-genomes have diverged after sub-population *δ* went silent ≈500 000 years ago. On one hand, this fits well with estimated B/G divergence of 900 000–500 000 years ago, based on chloroplast sequences[29]. On the other hand, it contradicts cytological data indicating a more ancient divergence[35]. One possible explanation for this discrepancy is that cross-species hybridizations have allowed *RLC_Angela β* sub-population elements to invade the *T. timopheevii* genome after its initial divergence from the B genome.

More puzzling are the introgressions in the A and D sub-genomes, which seem to come from donors that are similar to, but not identical with the A sub-genome. These introgressions differ in the more ancient *RLC_Angela δ* sub-population elements, while the younger *β* sub-population elements cluster with those from the wheat A sub-genome. It is possible that these represent A sub-genome relatives that diverged from the A sub-genome before or during the time when the *RLC_Angela δ* sub-population was most active. One can speculate that these evolutionary lineages were then geographically isolated, for example through glaciation during ice ages. After a few 100 000 years of separated evolution, these lineages could have been re-introgressed into the diploid or allotetraploid wheat gene pool, and populated by RLC_Angela *β* sub-population elements. Indeed, previous studies concluded that hybridizations and retic-

ulate evolution must have occurred multiple times during the evolution of the Triticeae[16,17] The A and D sub-genome introgressions described here could be remnants of such hybridization events.

TE population analysis also provided independent molecular dating of polyploidization and translocation events; we found that the major 7B/4A translocation must have occurred soon after the formation of allotetraploid wheat. Additionally, we identified TE sub-families that were active only in the diploid A and B genome progenitors. These all went largely silent about 400 000–600 000 years ago. Conversely, sub-families that spread across both the A and B sub-genomes (i.e., after allotetraploidization) became active around the same time. Thus, our retrotransposon insertion age estimates fit well with estimates from previous studies, despite them using different approaches to date divergence times of sub-genomes and polyploidization events[17,18] This indicates that the nucleotide substitution rate of 1.3E-8 per site per year[9] that we used for insertion age estimates yields results that are consistent with those obtained by other methods.

Interestingly, we did not find evidence for a broad "genomic shock" that was proposed to follow polyploidization events or introgression events.[36,37] Our data shows that allotetraploidization was followed by the activity of only a few retrotransposon sub-families that spread to both the A and B sub-genomes, while many others went silent. This finding is also consistent with our previous study that also found no evidence of TE activation following polyploidization.[8] Additionally, introgressions to chromosomes 2A and 6A contained many young and potentially active elements. However, we found no such young copies outside the introgressed segments, indicating that retrotransposons introduced in these events did not spread to the rest of the wheat genome. Based on these findings, we hypothesize that naturally occurring allopolyploids and large chromosomal introgressions can only survive if TE activity does not increase drastically, while studies reporting a "genomic shock" may describe young polyploids that would not be viable in the long term. Indeed, our data does not allow any conclusions as to the factors that trigger TE activity bursts. Long-term climate changes such as ice ages could, for example, be activating TEs. However, we also found no co-occurrence of activity of homologous families in the individual wheat sub-genome or in other Triticeae such as barely and rye (Figure S11, Supporting Information).

### 3.2. Non-Autonomous TEs Have Complex Partnerships with Autonomous Ones

An especially deep insight was gained into the co-evolution of autonomous and non-autonomous elements. The emergence of non-autonomous elements is not surprising from an evolutionary point of view. If a subset of TEs in a family lose a protein function, they can still replicate as long as TE copies are around that produce the functional protein. The *BARE2* element in barley may exemplify an early stage of such an evolution: its *GAG* gene is disrupted, while its other genes are still intact, and it presumably uses GAG proteins from *RLC_BARE1* elements[38] The non-autonomous *RLC_Angela* elements in wheat represent a next step with their entire CDS being degenerated. Importantly, our PCA and phylogenetic analyses of LTR sequences

strongly indicate that non-autonomous *RLC_Angela* sub-families are functionally closely linked and co-evolve with specific sub-families of autonomous elements. The *RLC_Angela* retrotransposons also show that non-autonomous elements can transition to new autonomous "hosts" if their original autonomous partners become silent. Such "host switches" could occur when, for example, LTR sequences are transferred between sub-populations via gene conversion or template switching during reverse transcription.[2,3]

The terminal evolutionary state may have been reached in the *RLG_Wilma*/*RLG_Sabrina*/*RLG_WHAM* (WSW) system, where sequence homology between autonomous and non-autonomous partners is reduced to a few essential motifs. If the three families ever evolved from a common ancestor, it was at least 35–45 million years ago, as evidenced by the presence of *RLG_WHAM* homologs in *Brachypodium*. Thus, non-autonomous elements apparently can rely for many million years on autonomous partners, even if they have diverged from them to a degree that they only share minimal sequence homology. Interestingly, the non-autonomous WSW retrotransposons described here differ from well-studied systems such as *Alu* elements which are mobilized by L1 retrotransposons.[39] *Alu* is not derived from autonomous L1 elements, but has instead evolved independently from 7SL RNA and is under the transcriptional control of its own internal RNA polymerase III promoter. The WSW elements are also fundamentally different from MITEs, which have no regulatory or coding sequences at all and are mobilized only because of short conserved motifs at their termini[10] Instead, WSW elements seem to all be part of a "cooperative complex" where different elements contribute specific proteins. The autonomous *RLG_Wilma* encodes the complete set of canonical proteins such as GAG, RT and INT, while the "semi-autonomous" elements *RLG_Sabrina* and *RLG_WHAM* encode GAG proteins, which are needed in large amounts to stabilize RNA intermediates in virus-like particles[40] Additionally, there are at least seven highly abundant *RLG_Sabrina* variants that contain unique CDS segments and which may also provide functional proteins. In summary, these data suggest that autonomous and non-autonomous TEs can co-evolve over long periods of time, and into highly complex systems. Furthermore, our finding of *RLG_WHAM* homologs in the small 270 Mb genome of *Brachypodium* suggests that a large and repetitive genome is not a prerequisite for the evolution of complex systems of autonomous and non-autonomous TEs. Future broad comparative surveys of multiple plant genomes would help answer the question on the evolutionary origin of the WSW retrotransposons.

## 4. Conclusions

With more and more large and repetitive genomes becoming available, population analysis of TEs can help answer a multitude of evolutionary questions. In this study, we only focused on a few families of LTR retrotransposons. It will be highly interesting to analyze TE populations of different super-families or classes, such as non-LTR retrotransposons or DNA transposons in future studies. Such analyses may also be useful to assess genetic diversity of species. For example, it is perceivable that large numbers of wheat (or other crop) lines are sequenced to a low coverage. These sequences could then be used to assemble high-

**2100022 (14 of 17)**

copy TEs to produce consensus sequences representing the main TE sub-families. In this way, lines that carry chromosomal segments from rare cultivars, land races or alien donors could be identified. This could help assess genetic diversity of crop species and assist breeding programs.

## 5. Experimental Section

*Software Sources*: Unless stated otherwise in the methods section, bioinformatics software was obtained from Ubuntu repositories (ubuntu.com).

*Identification and Isolation of Full-Length Retrotransposons*: The bioinformatics pipeline developed for this study identifies and isolates full-length TE copies belonging to single LTR retrotransposon families. Since sequence variation in individual TE families can be considerable, the first step was to isolate a few hundred LTR sequences from which (if necessary) multiple consensus LTRs were constructed to cover as much intra-family diversity as possible (Table 1): up to 100 LTRs were picked randomly and aligned with Clustalw using a gap opening penalty of 10 and a gap extension penalty of 0.2. From this first alignment, groups of LTR variants were identified by eye and selected for separate Clustalw alignments. From these, consensus sequences were constructed using the custom perl script consensus. These consensus LTRs were then used in blastn searches against wheat chromosomes. All blast hits that covered >95% of the length of the LTR and showed >80% identity were considered. In a second step, we checked for completeness by allowing a maximum of 5 bp missing on either side of the LTR. Full-length elements of a given TE family were identified by searching the blastn outputs for pairs of LTRs that were found in the same orientation at a distance that can be roughly expected based on the length of consensus sequences for the respective family. For example, for *RLC_Angela* elements the two LTRs had to be found within a range of 7700–9700 bp (a consensus *RLC_Angela* sequence has a length of ≈8700 bp). Subsequently, candidate full-length elements were screened by blastx for the presence of the expected polyprotein. Finally, copies that are at the extremes of the size distribution (e.g., the shortest and longest 3%) were discarded to eliminate TE copies with large insertions or deletions in order to avoid problems with large structural variations in the subsequent analyses

*Retrotransposon Insertion Age Estimates*: Insertion ages of individual retrotransposon were estimated by aligning the two LTRs with the program Water (EMBOSS package) using a gap opening penalty of ten and a gap extension penalty of 0.5. Nucleotide differences between LTRs were counted and transitions and transversions were distinguished for molecular dating as previously described[41] For all molecular dating, a rate of 1.3E-8 per site per year proposed for intergenic regions in grasses was used[9]. Molecular dating of insertions times was automated with the in-house Perl script date_pair.

*Principal Component Analysis*: All individual copies of a TE family were aligned with a consensus sequence of the respective family using the program Water using a gap opening penalty of 50 and a gap extension penalty of 0.1. Consensus sequences for TE families were obtained from the TREP database (https://www.botinst.uzh.ch/en/research/genetics/thomasWicker/trep-db.html). The central piece of the TEpop pipeline is an original Perl script pair_to_vcf that combines these pairwise alignments into a single variant call file (vcf) for the whole TE family population. Here, variants were used that occur in at least 10% of all copies (i.e., minor allele frequencies of 10%). Only nucleotide substitutions were considered, while insertion in retrotransposon copies were ignored, and deletions were treated as missing data, with a missing data cutoff at 90%. This script also integrates the insertion time for each TE copy. The vcf file was then be used principal component analyses (PCAs) using the R libraries gdsfmt, SNPRelate, ggplot2, and magrittr. The output table was used for visualization with the original Perl script visual_PCA_select_subfam. This script also allows sub-families to be defined by visual inspection using the PCA plot. This manual step is necessary since boundaries of sub-families are sometimes blurred. In such cases, only the central parts of sub-family

"clouds" were selected while the blurred region between subfamilies was excluded (examples in Figure S2b, Supporting Information). To construct consensus sequences of TE sub-families, 8–13 full length copies were picked randomly and aligned with Clustalw using a gap opening penalty of ten and a gap extension penalty of 0.2. Consensus sequences were constructed using the custom perl script consensus.

*Divergence Time Estimates of Retrotransposon Sub-Families and Sub-Populations*: To estimate the divergence time between sub-families or sub-population, a variation of our previously described molecular dating method was used[41,42] Coding sequences (CDS) were extracted by aligning each individual retrotransposon copy with a consensus sequence of the predicted CDS using the program Water. Individual TE copies that contain intact or near-intact ORFs (maximum three in-frame stop codons and no frame shifts) were selected from each sub-family or sub-population. Here, it was assumed that a CDS with three or less stop codons does not contain a frame shift. The predicted proteins of random pairs of TE copies from different sub-families/sub-populations were aligned with the Emboss program Water. The protein alignment was used to produce a codon-by-codon DNA alignment of the CDS for each pair. An estimate of the divergence time was obtained by considering only synonymous sites in four-fold degenerate sites. Here, the third positions of codons for Ala, Gly, Leu, Pro, Arg, Ser, Thr, and Val were used. For for Leu, Arg, and Ser (which all have six possible codons), only the codons starting with CT, TC, and CG, respectively, were used. Again, the substitution rate of 1.3E-8 per site was applied.

For the different divergence time estimates, between 43 and 478 gene pairs were used (the number was determined by the number of TE copies with near-intact ORFs). The mode (i.e., the peak) of the divergence time distribution was used as the estimate for the overall divergence time of sub-families and sub-populations.

*Phylogenetic Analyses*: DNA or protein sequences were aligned with CustalW using a gap opening penalty of 5.0 and a gap extension penalty of 0.01. Multiple alignments were converted to nexus format with Clustalx. Alignments were visually inspected for proper alignment of sequences. Phylogenetic trees were constructed with MrBayes using parameters lset nst = 6 rates = invgamma and the mcmc algorithm. BARE1 and BdC022, *RLC_Angela* homologs from barley and *Brachypodium distachyon* respectively, were used as outgroups to root the phylogenetic tree. The program was run for at least 100 000 generations, adding generations until the average standard deviation of split frequencies fell below 0.01. Phylogenetic trees were visualized with FigTree. Three-dimensional structural protein similarities were predicted with phyre2 (http://www.sbg.bio.ic.ac.uk/phyre2/).

*Analysis of Retrotransposon Expression Levels*: Previously published transcriptome datasets[32] of wheat infected with three different isolates of powdery mildew as well as infected control plants were used. The reads were mapped with the Salmon software version 1.5.2 (obtained from github.com/COMBINE-lab/salmon/releases) to a collection of consensus TE sequences compiled from the TREP database (https://www.botinst.uzh.ch/en/research/genetics/thomasWicker/trep-db.html). Mappings were run with default parameters. The calculated transcripts per million (TPM) values were used for the box plots.

## Supporting Information

Supporting Information is available from the Wiley Online Library or from the author.

## Acknowledgements

H.G. and N.S. jointly supervised this work. This work was supported by core funding of the University of Zurich, by the Swiss National Foundation grant 31003A_163325 and by the University Research Priority Program grant U-702-21-01 of the University of Zurich. The project was made possible through data generated in the framework of the wheat 10+ Genome Project.

## Conflict of Interest

The authors declare no conflict of interest.

## Data Availability Statement

Data openly available in a public repository that does not issue DOIs. All custom Perl scripts used for this study were deposited at github (https://github.com/wicker314/TEpop).

## Peer Review

The peer review history for this article is available in the Supporting Information for this article.

## Keywords

chromosomal introgression, LTR-retrotransposon, non-autonomous element, TE population

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

www.advancedsciencenews.com

ADVANCED
GENETICS

www.advgenet.com

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
