## [**Supplementary Information**: Record of Transparent Peer Review · Advanced Genetics]

Record of Transparent Peer Review

Transposable element populations shed light on the evolutionary history of wheat and the complex co-evolution of autonomous and non-autonomous retrotransposons

Thomas Wicker*, Christoph Stritt, Alexandros G. Sotiropoulos, Manuel Poretti, Curtis Pozniak, Sean Walkowiak, Heidrun Gundlach, Nils Stein

*Corresponding

Review timeline:

Date Submitted: 03-Jun-2021
Editorial Decision: 09-Jul-2021 Minor Revision
Revision Received: 31-Aug-2021
Accepted: 03-Sep-2021

Editor: Myles Axton

Initial Editorial Evaluation	04-Jun-2021
-------------

Summary

Transposon families from 10 high quality wheat genomes were analyzed with respect to their historical activity periods, subgenome localization and relationship to chromosomal rearrangements.

Scope

This kind of data analysis documenting the allohexaploid history of one of our most important staple crops raises interesting questions about the external factors that correlated with bursts of transposon activity. The internal population evolution among repeat families is a fascinating research area in its own right. This is both a resource of well characterized markers and a fertile research area in its own right.

1 st Peer Review	04-Jun to 30-Jun-2021
-----------------------

Reviewer #1

This work has carried a detailed analysis of a few – but high-copy—LTR retrotransposons families and derived sub families. They have determined the evolutionary history of these TEs families, dating waves of amplification during wheat evolution and studying the co-evolution of autonomous vs. non autonomous elements. They also used these TEs as using markers to analyze translocations or homoeologous recombination events, or introgressions. The novelty of the work can be summarized as follows:

-Using new high quality whole genome data, that has been published by the authors in the pan-genome study on 10 de novo sequenced wheat lines (Walkowiak et al., 2020), to perform a detailed analysis on 5 selected LTR retroelements families and classifying these elements as autonomous versus non-autonomous; and the dating TEs activity based on LTRs sequence divergence.

-Using independent dating approach compared to previous studies analyzing base substitutions, to address some important milestones in wheat evolution (formation of tetraploid wheat or timing of D genome hybridization or divergence of B and G genomes. This enabled analysis of TE activity dating relative to wheat evolution milestones mentioned above.

-Showing co-evolution of autonomous and non-autonomous elements sub-families, with a specific case-study showing a non-obvious connection between TE Wilma (as autonomous element) that is a candidate for activation of non-autonomous Sabrina and WHAM TEs.

Overall, the analyses presented are of high quality. The most original aspect of the work is the identification of co-evolving autonomous and non-autonomous families. Other parts are expanding our knowledge on TEs evolutionary history in wheat.

Comments:

1.1-It would have been refreshing if the authors would have been willing to risk themselves a bit more on mechanistic hypotheses to explain what they see. For example, the authors could address what might correlate with bursts of transposition. They mention that allotetraploidization between A and B did not trigger TE bursts --even-though they report it did activate a few families. What about all other cases of introgressions? They also start by wide hybridizations (even if not associated with genome doubling)-- Such wide crosses were shown in wheat and Arabidopsis and tomato to cause dysregulation of gene activity in general including TE activities. What about climate change events? This of course might be a bit speculative but if similar dating of waves of amplification occurred independently in other species (e.g. Barley, Rye, Brachypodium etc..) this would be quite interesting. The authors have also published rye and barley genomes so they might have insight on that.

1.2- Another point that might have spiced the article would be to try and provide a Proof of Concept for the activity of some elements in real time. For example, in wheat lines where the authors predict the presence of potentially active autonomous and non-autonomous elements, they could test, by one of the transposon-display methods, whether they can detect de-novo activity of these TEs.

1.3- One statement at the bottom of page 14 does not make sense:

“B and G sub-genomes must have diverged after sub-population δ went silent approximately 500,000 years ago.”

B and G divergence is much older than that.... B diverged from *Aegilops speltoides* 2-3 Myrs ago while G is closer to *Ae. speltoides*. So it cannot be that B and G diverged after *Triticum dicoccoides* formation ~ 800,000 years ago. Check your literature on that. What is possible is that *dicoccoides* and *timopheevii* did hybridize quite frequently which could explain some of the data.

1.4-There are hints in the text, but I was wondering if there is no evidence for invasion of A or B elements in D and vis versa since hexaploid wheat formation. Maybe this can be stated more clearly. Along the same line the authors say – “very young elements may still be missing from the current assemblies”. This sentence is not clear enough. How young is “very young”. In other terms what is the resolution of the analysis when determining transposition in the past 10,000 years, e.g. after hexaploid wheat formation?

1.5- Characterization of introgressions/rearrangements-- The use of TEs to determine introgressions is not new. Moreover, introgressions were already mapped in the (Walkowiak et al., 2020) publications. Please clarify what you have learned here, using TEs, that was not known already or that could not be determined by SNPs analysis.

1.6- in Figure 3; please explain better what you concluded from comparing the PCA of the different Angela populations.

1.7- Results, page 8 – It is worth mentioning that Cereba family is also present today in the genomes of wheat diploid species (*T.urartu*, *T.monococcum*, *Ae.tauschii*, *Ae.speltoides*) as well as in the barley and rye genomes. Was this family also present in the Triticeae ancestor, like the Angela family?

Minor comments

1m1. Page 8, line 3 : “ Subfamilies where defined” – correct to “were”

1m2. Page 8, line 4 : “when boundaries between the are blurred” – correct sentence

1m3. Page 8, line 9 : “ancestor of the A, B and sub-genomes” - missing “D”

1m4. Figure 1 legend, line 5: “D1 and D2 are are” – remove one “are”.

Line 10 : “Divergence divergence”

1m5. Fix typo at the top of Page 9: “when boundaries between the are blurred (Fig.1b).”

1m6. Abstract, line 13: “Tes” should be “TEs”

1m7. Page 10, line 18: “introgression is was shown” - please correct sentence.

1m8. Page 14, 2nd paragraph: “since recombination between sub-genomes are”, please correct sentence

1m9. Page 14, 3rd paragraph: “polyppoidization” should be “polyploidization”

1m10. Methods, page 16: “within an range”, ‘an’ should be ‘a’

1m11. Supplemental table 2: “were used if if they” – delete one if

Reviewer #2

In this manuscript, Wicker et al. investigated the sequence structure, genomic distribution, divergence time, proliferating mechanisms of five LTR retrotransposon families in wheat. Under a TE-population framework the authors discovered potential values of using TE polymorphisms to date polyploidization events and detecting introgression segments. The analytic methods used are sound, the presentation and interpretation of results is logical and valid. The finding that non-autonomous TE families utilize components of autonomous/semi-autonomous TE families to propagate, and together they co-evolve for 30-40 million years is intriguing. I have only one major concern and several minor complaints, which are described in detail below:

2.1 The details of the TE-population framework should be described: how consensus sequences of each TE sub-family is constructed, what parameters were used to align each TE copy to the sub-family consensus and how variants were identified, what types of variants (SNPs, indels), minor allele frequency, missing data cutoff, etc. were used to filter the VCF, how are large structural variants handled?

2.2 Also, how does removal of structurally incomplete TEs and TEs of extreme size affect the PCA results and dating estimates?

minor comments:

2m1. supplemental figure 3: The regression lines in all three lines intercept y-axis at near 0 position, which are inconsistent with the reported intercepts (0.18-0.42).

2m2. Page 13 line 2: Fig. 6e should be Fig. 7e.

2m3. This is totally a personal request, I'd really like to hear some discussion on the cross-species prevalence of the non-autonomous/autonomous/semi-autonomous co-evolving system. Is it specific to wheat and barley? Does it have anything to do with the extreme genome size of wheat and barley?

Reviewer #3

Wicker et al. present a detailed population-level survey of the highly abundant LTR retrotransposon families of the wheat genome. This approach is novel, and to my knowledge hasn't been applied with genome-scale data. Perhaps most interesting is the ability to track introgression from distinct lineages using TEs active at different times. As noted in the conclusion, the paper is of most interest to a specialist in wheat and transposable elements, due to the level of detail presented. It's almost a “natural history” of the wheat genome! Stylistically, there's a lot of interpretation in the results. Not sure on the journal requirements, but if possible merge results and discussion into one section? Below are a few major overarching comments, then comments that somewhat follow the narrative of the paper (I felt this was best because there unfortunately aren't line numbers on the manuscript I read).

3.1 The github repository for the paper is empty, so I had a really hard time assessing how the variants were generated. After aligning individual copies to the consensus, what happens to indels? Do these come out in your VCF? If there's an insertion of 4 bp in one individual and an insertion of 2 bp in another individual, how are these integrated in the VCF? What about if there's an insertion of 4 bp in one sample and an insertion of 4 bp in another sample with a SNP variant within the insertion? If it's the case that indels are ignored, how do you end up with frameshifts?

3.2 Throughout the paper, “wheat haplotype” is used to describe what I would call genotypes or cultivars. It's a little jarring to hear that a TE is active in a particular haplotype, since that's not really what's happening. To me, activity is jumps occurring within the nucleus of a germ cell of an individual. I'm not familiar enough with wheat nomenclature to know if this is a specialized term used to define something other than a contiguous section of sequence not disrupted by recombination (what I'd call a haplotype), but I'd suggest changing to genotype or cultivar for a general audience.

3.3 Background, page 4 middle: It's written that "The LTRs are typically 1-2 kb long," referring to all plant LTR retrotransposons - that's not true of many plant species, where sometimes a majority of LTRs are <1000 bp, e.g. Figure 5 in Jedlicka, Lexa, and Kejnovsky (2020) *Frontiers in Plant Science* <https://www.frontiersin.org/articles/10.3389/fpls.2020.00644/full#F5>

3.4 The length of the TE identified is confusing to me, e.g. page 6 where "LTRs were found by chance in the same orientation and at the right distance." What does the right distance mean? In the methods, when ranges are provided, they're not centered on the consensus distance (e.g. 7800-9300, but the consensus is 8700 bp, so 900 bp shorter and 600 bp longer?). Are these the approximate lengths remaining after chopping 3% outlier tails of length?

3.5 I disagree with the last paragraph of the dead on arrival section on page 7. The argument is made that low GC content means A/T rich stop codons are emerging through mutations more commonly. Since transition mutations are more common than transversion mutations, as well as the high rate of deamination of methylated cytosines to T, I think high GC content would make stop codons more likely to emerge. Can the authors describe their logic here?

3.6 When barley is introduced on page 8, it would be helpful to know how diverged in Mya wheat and barley are, to understand how the BARE1 homology can be expected.

3.7 Page 10: When the G genome is introduced, I thought it was a typo. How G relates to A, B, and D needs to be introduced for a non-wheat audience.

3.8 On the enrichment in introgressions, I buy the argument, but the pattern could also be generated by a different process. I assume regions of introgression are in regions with lower recombination, and thus lower selection efficacy. If TEs are effectively removed elsewhere in the genome, this pattern might be expected.

3.9 With the Sabrina section, the internal proteins need more explanation. Even looking at figure 5, it's unclear what these short proteins are. What is a VLP gene? Is it GAG since that's generating the capsid? Do these proteins blast to genes of non-TE origin, or are they all deletion derivatives of Sabrina?

3.10 Why do wheat homologs have a different name for some families but not all (BAGY2/Wilma being different vs. Sabrina or WHAM which seem to be the same in barley and wheat)? Is there higher sequence divergence of the autonomous copies than the nonautonomous? Although the BAGY2 presence in barley is described in the introduction, by the time I got to the results I had forgotten, and it's worth reiterating there.

3.11 The five criteria presented for autonomous/nonautonomous families could be defined better. The first, "PCAs of autonomous and nonautonomous families should look similar" is tricky because what does "look similar" really mean? To me, a real test would be whether frameshift or premature stop codons are loaded on the PC that differentiates autonomous from nonautonomous families. That would be an actual falsifiable criteria for suggesting these are really coevolving, and also comparable across different families (the proportion of all frameshifts/stops that are loaded on the PC separating autonomous from nonautonomous). Also, the third requirement that the PBS needs the same tRNA primer doesn't seem true - is there evidence that RT needs anything other than a free 3' RNA end to get going? I get that there's a lot of stasis in PBS, as there's lots of nucleotides to change to evolve away to a brand new tRNA, but I don't think it's necessarily a requirement for autonomous/nonautonomous families.

3.12 In the methods: More information about things like blast parameters, how the range of lengths was determined (it's not symmetric around the consensus length), what the expected polyprotein sequence is, are needed for reproducibility.

3.1b I was excited to see it written that the scripts for this study were available on github, but the repository is empty. It would have been a whole lot easier as a reviewer if they were available!

3.13 Figures:

PC percentages loaded on x and y would be helpful - if something like 95% of the variation is explained by the first two PCs, it would make it clear whether there's more separation of "sub-populations" on PC3 that would separate them out to looking like "sub-families".

Minor comments (sorry, no line numbers on the proof!):

3m1 Abstract: Third to last paragraph Tes isn't capitalized properly

3m2 Results: Page 6, principle should be principal in PCA

3m3 Figure 5 legend: typo, Poteinase

3m4 Supp Fig 3 needs a legend for what bubble size corresponds to what # of elements

1 st Editorial Decision	09-Jul-2021
-------------

Editorial decision: resubmit with Minor Revision

Editor’s understanding of the reviews

Reviewer #1 Recommends Minor Revision

Reviewer #2 Recommends Minor Revision

Reviewer #3 Recommends Major Revision

These are the main reviewer recommendations that the editors believe will make the biggest improvement to this article. **Please do address all reviewer comments listed in the decision letter in your point-by-point response** (you may continue this table to do so if you wish). We hope this summary helps you to understand our decision and expedites the revision process. We value feedback from author and referees alike.

Reviewer comments	Editor recommendation
3.1 The github repository for the paper is empty, so I had a really hard time assessing how the variants were generated	ED1 Please make the code and data available to the reviewers as a condition of resubmission.
2.1 The details of the TE-population framework should be described: how consensus sequences of each TE sub-family is constructed, what parameters were used to align each TE copy to the sub-family consensus and how variants were identified, what types of variants (SNPs, indels), minor allele frequency, missing data cutoff, etc. were used to filter the VCF, how are large structural variants handled? 3.12 In the methods: More information about things like blast parameters, how the range of lengths was determined (it’s not symmetric around the consensus length), what the expected polyprotein sequence is, are needed for reproducibility. 3.4 The length of the TE identified is confusing to me 3.3 Background: “The LTRs are typically 1-2 kb long,” referring to all plant LTR retrotransposons - that’s not true of many plant species, where sometimes a majority of LTRs are <1000 bp 2.2 Also, how does removal of structurally incomplete TEs and TEs of extreme size affect the PCA results and dating estimates?	ED2 these recommendations greatly increase the reusability of the approach and hence its impact. Please specify the lengths expected from prior literature. Detail the methods used to establish consensus or TE-specific sequence signatures, and how the individual instances of members of each family were measured in the sequence assemblies. Make clear how the filtering of repeats affects the results and conclusions.
1.1 what might correlate with bursts of transposition?	ED3 I would be open to more speculation based on comparison with other allopolyploid genomes (reproducible negative) or rye and barley genomes (convergent effect of climate or agricultural selective pressure).
3.11 ...a real test would be whether frameshift or premature stop codons are loaded on the PC that differentiates autonomous from nonautonomous families. That would be an	ED4 I think the in silico genetic test of autonomous versus nonautonomous families using the loading of lof mutations on the PC would be appropriate way to test the

actual falsifiable criteria for suggesting these are really coevolving, and also comparable across different families (the proportion of all frameshifts/stops that are loaded on the PC separating autonomous from nonautonomous) 1.2 in wheat lines where the authors predict the presence of potentially active autonomous and non-autonomous elements, they could test, by one of the transposon-display methods, whether they can detect de-novo activity of these TEs.	criteria in this analysis paper. However, I would not be averse to the inclusion of existing transposon display data if that might be informative. If there is an available transposon display experiment that can be re-analyzed, can the results be predicted from your analysis of active and non-autonomous elements? Can retrotransposon activity be identified by differential transcription of nearby genes?
2m3 I'd really like to hear some discussion on the cross-species prevalence of the non-autonomous/autonomous/semi-autonomous co-evolving system. Is it specific to wheat and barley? Does it have anything to do with the extreme genome size of wheat and barley?	ED5 Since the mix of autonomous and non-autonomous elements has been reported in various systems, across grasses or agricultural plant species, is there a numerical relationship between genome size or ploidy and the proportions of each element type, or is this just a function of time?
3.2 Throughout the paper, "wheat haplotype" is used to describe what I would call genotypes or cultivars.	ED6 The haplotype is strictly the sequence of genetic variants linked in cis with the transposon studied rather than a label for the plant accession in which the transposon and haplotype were studied.

Author's Response to 1st Review

31-Aug-2021

Reviewer comments	Editor recommendation	Author reply
3.1 The github repository for the paper is empty, so I had a really hard time assessing how the variants were generated	ED1 Please make the code and data available to the reviewers as a condition of resubmission.	We apologize for the empty github repository. This was an oversight. We had all scripts ready, but simply forgot the deposit them. All scripts and necessary instruction are now deposited at https://github.com/wicker314/
2.1 The details of the TE-population framework should be described: how consensus sequences of each TE sub-family is constructed, what parameters were used to align each TE copy to the sub-family consensus and how variants were identified, what types of variants (SNPs, indels), minor allele frequency, missing data cutoff, etc. were used to filter the VCF, how are large structural variants handled? 3.12 In the methods: More information about things like blast parameters, how the range of lengths was determined (it's not symmetric around the consensus length), what the expected polyprotein sequence is, are needed for reproducibility. 3.4 The length of the TE identified is confusing to me 3.3 Background: "The LTRs are typically 1-2 kb long," referring to all plant LTR	ED2 these recommendations greatly increase the reusability of the approach and hence its impact. Please specify the lengths expected from prior literature. Detail the methods used to establish consensus or TE-specific sequence signatures, and how the individual instances of members of each family were measured in the sequence assemblies.	2.1. We have added detailed descriptions on alignment softwares and parameters that were used to construct consensus sequences (page 16, paragraph 1 and page 17, paragraph 1). Furthermore, we added information on minor allele frequencies and missing data cutoff used for variant calling (page 16, paragraph 1). 3.12. We have added the information to the methods section on parameters for blast searches, pairwise and multiple alignments and the production of the variant call file (pages 16-18, see also our response to comment 2.1).

retrotransposons - that's not true of many plant species, where sometimes a majority of LTRs are <1000 bp 2.2 Also, how does removal of structurally incomplete TEs and TEs of extreme size affect the PCA results and dating estimates?	Make clear how the filtering of repeats affects the results and conclusions.	3.4. We have made the statement more precise stating that the LTRs have to be approximately at the distance that is expected for the respective family (Page 5, paragraph 2). 3.3. The reviewer is correct. Even though in wheat, LTRs are usually longer, most are between 500 and 2000 bp. We made the statement more specific, mentioning LTRs are typically 0.5-2 kb in wheat and other Triticeae (Page 3, paragraph 2). This had no influence on results or conclusions. 2.2 We also added a statement to the methods emphasizing that copies with large insertions and deletions (i.e. large structural variants) were excluded (page 16, paragraph 1). The reason for this exclusion is the following: a majority of TEs in wheat are fragmented by deletions or through insertion of (sometimes numerous) other TEs. This makes it technically extremely challenging to process such copies, as precise borders of deletions and insertions have to be determined with a resolution of maximum a few bp to avoid introducing sequence polymorphism artifacts. Thus, we decided to focus only on full-length elements with only small InDels. This is a clear limitation of the study, as many older TE copies tend to be fragmented and thus are not considered in this study. We have added statements in the introduction (page 3, paragraph 2) and results (page 5, paragraph 1) making this limitation clear.
1.1 what might correlate with bursts of transposition?	ED3 I would be open to more speculation based on comparison with other allopolyploid genomes (reproducible negative) or rye and barley genomes (convergent effect of climate or agricultural selective pressure).	1.1 We compared insertion ages of two TE families (RLC_Angela and RLG_Wilma) between wheat sub-genomes and with their homologs in barley and rye. We found no co-occurring bursts indicative of an effect of climate in sub-genomes or between species (except for the described activity of a few families

		after tetraploidization). We show these results in the new Supplementary Figure 12 and have added a few sentences discussing this topic (Page 15, paragraph 1).
3.11 ...a real test would be whether frameshift or premature stop codons are loaded on the PC that differentiates autonomous from nonautonomous families. That would be an actual falsifiable criteria for suggesting these are really coevolving, and also comparable across different families (the proportion of all frameshifts/stops that are loaded on the PC separating autonomous from nonautonomous) 1.2 in wheat lines where the authors predict the presence of potentially active autonomous and non-autonomous elements, they could test, by one of the transposon-display methods, whether they can detect de-novo activity of these TEs.	ED4 I think the in silico genetic test of autonomous versus nonautonomous families using the loading of lof mutations on the PC would be appropriate way to test the criteria in this analysis paper. However, I would not be averse to the inclusion of existing transposon display data if that might be informative. If there is an available transposon display experiment that can be re-analyzed, can the results be predicted from your analysis of active and non-autonomous elements? Can retrotransposon activity be identified by differential transcription of nearby genes?	For both, the RLC_Angela and the Wilma/WHAM/Sabrina system, the non-autonomous elements are so far diverged from the autonomous ones that CDS can simply not aligned in any meaningful way anymore. In fact, loading RLG_Wilma, WHAM and Sabrina into the same PCA would not work due to the near complete lack of sequence conservation. 1.2 We agree with the reviewer that it would be very nice to have experimental data. Unfortunately, we currently do not have the resources, and such experiments would also be too time-consuming to perform for this revision. Because also other reviewers' comments bring up the topic of recent retrotransposon activity (see below), we performed a series of additional analyses: (i) we searched for TE copies with identical LTRs (i.e. inserted too recently to accumulate any mutations) which are present in only one wheat genotype and for which we could find the exact orthologous region in other wheat genomes that do not contain the insertion, (ii) we performed phylogenetic analyses to show transposition of A/B genome retrotransposons to the D genome after hexaploidization, and (iii) we searched transcriptome data for evidence of transcription. The results of these analyses are described in an additional paragraph at the end of the results section (starting page 12, paragraph 3) and shown in the new supplementary figure 12. We also added the corresponding methods (page 18, paragraph 2). The main findings are that we indeed find some evidence of very recent or current activity, but the five retrotransposon families studied here seem to be

		mostly silent after having long periods of very high activity during the past 2-3 million years.
2m3 I'd really like to hear some discussion on the cross-species prevalence of the non-autonomous/autonomous/semi-autonomous co-evolving system. Is it specific to wheat and barley? Does it have anything to do with the extreme genome size of wheat and barley?	ED5 Since the mix of autonomous and non-autonomous elements has been reported in various systems, across grasses or agricultural plant species, is there a numerical relationship between genome size or ploidy and the proportions of each element type, or is this just a function of time?	2.3 This is a highly interesting question to which, at the moment, we have no definitive answer. Highly abundant non-autonomous elements are found in almost all organism groups (e.g. Alu in human, MITES in all grasses, etc.). However, such a complex system of non-autonomous/autonomous/semi-autonomous co-evolving elements was new to us. The fact that we found (although very small numbers of) RLG_WHAM homologs in Brachypodium suggests that such systems can prevail also in small genomes. We have added a few statements discussing this point to the discussion (Page 15, paragraph 1).
3.2 Throughout the paper, "wheat haplotype" is used to describe what I would call genotypes or cultivars.	ED6 The haplotype is strictly the sequence of genetic variants linked in cis with the transposon studied rather than a label for the plant accession in which the transposon and haplotype were studied.	3.2 The reviewer's definition of haplotype is correct. This term is often (and incorrectly) used as "lab slang" in the wheat community, which is why it was used so prevalent in our manuscript. We changed it throughout the manuscript it to more appropriate terms such as "wheat lineages", "wheat lines", "cultivars, "evolutionary lineages" or "sub-genome relatives", where necessary.

Reviewer(s)' Comments to Author:

Reviewer: 1

Comments to the Author

This work has carried a detailed analysis of a few – but high-copy—LTR retrotransposons families and derived sub families. They have determined the evolutionary history of these TEs families, dating waves of amplification during wheat evolution and studying the co-evolution of autonomous vs. non autonomous elements. They also used these TEs as using markers to analyze translocations or homoeologous recombination events, or introgressions. The novelty of the work can be summarized as follows:

-Using new high quality whole genome data, that has been published by the authors in the pan-genome study on 10 de novo sequenced wheat lines (Walkowiak et al., 2020), to perform a detailed analysis on 5 selected LTR retroelements families and classifying these elements as autonomous versus non-autonomous; and the dating TEs activity based on LTRs sequence divergence.

-Using independent dating approach compared to previous studies analyzing base substitutions, to address some important milestones in wheat evolution (formation of tetraploid wheat or timing of D genome hybridization or divergence of B and G genomes. This enabled analysis of TE activity dating relative to wheat evolution milestones mentioned above.

-Showing co-evolution of autonomous and non-autonomous elements sub-families, with a specific case-study showing a non-obvious connection between TE Wilma (as autonomous element) that is a candidate for activation of non-autonomous Sabrina and WHAM TEs.

Overall, the analyses presented are of high quality. The most original aspect of the work is the identification of co-evolving autonomous and non-autonomous families. Other parts are expanding our knowledge on TEs evolutionary history in wheat.

Comments:

Reviewer's comment 1.1

1-It would have been refreshing if the authors would have been willing to risk themselves a bit more on mechanistic hypotheses to explain what they see. For example, the authors could address what might correlate with bursts of transposition. They mention that allotetraploidization between A and B did not trigger TEs bursts --even-though they report it did activate a few families. What about all other cases of introgressions? They also start by wide hybridizations (even if not associated with genome doubling)-- Such wide crosses were shown in wheat and Arabidopsis and tomato to cause dysregulation of gene activity in general including TEs activities. What about climate change events? This of course might be a bit speculative but if similar dating of waves of amplification occurred independently in other species (e.g. Barley, Rye, Brachypodium etc..) this would be quite interesting. The authors have also published rye and barley genomes so they might have insight on that.

Our response

The reviewer raises an interesting point which we addressed with a new analysis of *RLC_Angela* retrotransposons of the $\delta 1a$ and $\delta 1n$ subfamilies (where the high number of retrotransposons allowed a high resolution analysis). We found that the introgressed segments enriched in these sub-families also contain many much younger copies than the rest of the wheat genome, indicating that they were active in more recent times in the donor species. Furthermore, this indicates that the $\delta 1a$ and $\delta 1n$ subfamily elements did not spread much (or not at all), since the rest of the wheat genome contains no $\delta 1a$ and $\delta 1n$ insertions of such young age. We added this information to the results section (page 8, paragraph 3) and show the data in the new Supplementary Figure 8. We also expanded the discussion of this topic, as suggested by the reviewer (page 15, paragraph 1).

Additionally, in response to the reviewer's comment, we compared insertion ages of two TE families (*RLC_Angela* and *RLG_Wilma*) between wheat sub-genomes and with their homologs in barley and rye. We found no co-occurring bursts indicative of an effect of climate in sub-genomes or between species (except for the described activity of a few families after tetraploidization). We show these results in the new Supplementary Figure 12 and have added a few sentences discussing this topic (Page 15, paragraph 1).

Reviewer's comment 1.2

2- Another point that might have spiced the article would be to try and provide a Proof of Concept for the activity of some elements in real time. For example, in wheat lines where the authors predict the presence of potentially active autonomous and non-autonomous elements, they could test, by one of the transposon-display methods, whether they can detect de-novo activity of these TEs.

Our response

We agree with the reviewer that it would be very nice to have experimental data. Unfortunately, we currently do not have the resources, and such experiments would also be too time-consuming to perform for this revision. Because also other reviewers' comments bring up the topic of recent retrotransposon activity (see below), we performed three additional analyses to specifically address these comments: (i) we searched for TE copies with identical LTRs (i.e. inserted too recently to accumulate any mutations) which are present in only one wheat genotype and for which we could find the exact orthologous region in other wheat genomes that do not contain the insertion, (ii) we performed a phylogenetic analysis of 99 recently inserted *RLC_Angela* elements to show transposition of A/B genome retrotransposons to the D genome after hexaploidization, and (iii) we searched transcriptome data for evidence of transcription. The results of these analyses are described in an additional short paragraph at the end of the results section (starting page 12, paragraph 3) and shown in the new supplementary figure 12. We also added the corresponding methods (page 18, paragraph 2). The main findings are that we indeed find some evidence of very recent or current activity, but the five retrotransposon families studied here seem to be mostly silent after having long periods of very high activity during the past 2-3 million years.

Reviewer's comment 1.3

3- One statement at the bottom of page 14 does not make sense:

"B and G sub-genomes must have diverged after sub-population δ went silent approximately 500,000 years ago."

B and G divergence is much older than that.... B diverged from *Aegilops speltoides* 2-3 Myrs ago while G is closer to *Ae. speltoides*. So it cannot be that B and G diverged after *Triticum dicoccoides* formation ~ 800,000 years ago. Check your literature on that. What is possible is that *dicoccoides* and *timopheevii* did hybridize quite frequently which could explain some of the data.

Our response

This is indeed an apparent contradiction. However, the literature is, to our knowledge, not completely clear on this, as Gornicki et al., 2014 suggest a quite recent B/G divergence of 0.9-0.5 million years. We have now formulated the claim more cautiously and added an additional citation (Ruban & Badaeva, 2018) and a short discussion that includes the possibility of hybridizations between species (Page 13, paragraph 1).

Reviewer's comment 1.4

4-There are hints in the text, but I was wondering if there is no evidence for invasion of A or B elements in D and vis versa since hexaploid wheat formation. Maybe this can be stated more clearly. Along the same line the authors say – "very young elements may still be missing from the current assemblies". This sentence is not clear enough. How young is "very young" . In other terms what is the resolution of the analysis when determining transposition in the past 10,000 years, e.g. after hexaploid wheat formation?

Our response

We have added a clarifying statement that "very young" elements are those with identical or near-identical LTRs. Their LTRs sometimes contain sequence gaps (stretches of N's) and are therefore not present as complete un-interrupted sequences (page 6, paragraph 1).

Additionally, we performed a phylogenetic analysis of the youngest copies of the most recently active *RLC_Angela* sub-population (β) to search for evidence of transposition from the A or B to the D subgenome (and vice versa). Indeed, we identified 2 clades containing 6 D-genome copies that descended from A/B genome elements, indicating that there was indeed transposition after hexaploidization. We have added these results to the new paragraph (starting page 12, paragraph 3, see also response 1.2 above) and show the tree in the new Supplementary Figure 12.

Reviewer's comment 1.5

5- Characterization of introgressions/rearrangements-- The use of TEs to determine introgressions is not new. Moreover, introgressions were already mapped in the (Walkowiak et al., 2020) publications. Please clarify what you have learned here, using TEs, that was not known already or that could not be determined by SNPs analysis.

Our response

The Walkowiak study simply identified the location and approximate boundaries of introgressed segments based on unique polymorphic TEs, but did not attempt to further characterize the possible origins of the introgressed segments. Here, we used TE population analyses to identify possible donor species (or species groups). The most remarkable finding was that some introgressions originate from relatives of different sub-genome donors. This type of information can not be gained from SNP analysis if the donor species is not known (or no sequence data is available). We added statements emphasizing this point to the discussion (Page 13, paragraph 3).

Reviewer's comment 1.6

6- in Figure 3; please explain better what you concluded from comparing the PCA of the different Angela populations.

Our response

We have added conclusions statements for each panel of Figure 3 to the figure legend.

Reviewer's comment 1.7

7- Results, page 8 – It is worth mentioning that Cereba family is also present today in the genomes of wheat diploid species (T.urartu, T.monococcum, Ae.tauschii, Ae.speltoides) as well as in the barley and rye genomes. Was this family was also present in the Triticeae ancestor, like the Angela family?

Our response

In fact, all five retrotransposon families used for this analysis are found in all known Triticeae genomes, and it is generally assumed that they were present already in the Triticeae ancestor. We added a statement emphasizing this point in the very beginning of the results section (Page 4, paragraph 1).

Reviewer's comment 1.8

Minor comments

1. Page 8, line 3 : “ Subfamilies where defined” – correct to “were”
2. Page 8, line 4 : “when boundaries between the are blurred” – correct sentence
3. Page 8, line 9 : “ancestor of the A, B and sub-genomes” - missing “D”
4. Figure 1 legend, line 5: “D1 and D2 are are” – remove one “are”.
Line 10 : “Divergence divergence”
5. Fix typo at the top of Page 9: “when boundaries between the are blurred (Fig.1b).”
6. Abstract, line 13: “Tes” should be “TEs”
7. Page 10, line 18: “introgression is was shown” - please correct sentence.
8. Page 14, 2nd paragraph: “since recombination between sub-genomes are”, please correct sentence
9. Page 14, 3rd paragraph: “polyppoidization” should be “polyploidization”
10. Methods, page 16: “within an range”, ‘an’ should be ‘a’
11. Supplemental table 2: “were used if if they” – delete one if

Our response

All corrections were made as indicated

Reviewer: 2

Comments to the Author

In this manuscript, Wicker et al. investigated the sequence structure, genomic distribution, divergence time, proliferating mechanisms of five LTR retrotransposon families in wheat. Under a TE-population framework the authors discovered potential values of using TE polymorphisms to date polyploidization events and detecting introgression segments. The analytic methods used are sound, the presentation and interpretation of results is logical and valid. The finding that non-autonomous TE families utilize components of autonomous/semi-autonomous TE families to propagate, and together they co-evolve for 30-40 million years is intriguing. I have only one major concern and several minor complaints, which are described in detail below:

Reviewer's comment 2.1

The details of the TE-population framework should be described: how consensus sequences of each TE sub-family is constructed, what parameters were used to align each TE copy to the sub-family consensus and how variants were identified, what types of variants (SNPs, indels), minor allele frequency, missing data cutoff, etc. were used to filter the VCF, how are large

structural variants handled? Also, how does removal of structurally incomplete TEs and TEs of extreme size affect the PCA results and dating estimates?

Our response

We have added detailed descriptions on alignment softwares and parameters that were used to construct consensus sequences (page 16, paragraph 1 and page 17, paragraph 1). Furthermore, we added information on minor allele frequencies and missing data cutoff used for variant calling (page 16, paragraph 1).

We also added a statement to the methods emphasizing that copies with large insertions and deletions (i.e. large structural variants) were excluded (page 16, paragraph 1). The reason for this exclusion is the following: a majority of TEs in wheat are fragmented by deletions or through insertion of (sometimes numerous) other TEs. This makes it technically extremely challenging to process such copies, as precise borders of deletions and insertions have to be determined with a resolution of maximum a few bp to avoid introducing sequence polymorphism artifacts. Thus, we decided to focus only on full-length elements with only small InDels. This is a clear limitation of the study, as many older TE copies tend to be fragmented and thus are not considered in this study. We have added statements in the introduction (page 3, paragraph 2) and results (page 5, paragraph 1) making this limitation clear.

minor comments

Reviewer's comment 2.2

supplemental figure 3: The regression lines in all three lines intercept y-axis at near 0 position, which are inconsistent with the reported intercepts (0.18-0.42).

Our response

For *RLC_Angela* and *RLG_Cereba*, the reported intercept points are actually correct. It may be slightly confusing that the y-axis labels are shifted a bit to the left to allow the display of data points with x values of zero. We added a clarifying statement to the figure legend.

However, for *RLG_Wilma*, the reported values were indeed incorrect, as we mistakenly used preliminary and not matching datasets for the "bubble" plot and the linear regression. We re-did the calculations which resulted in a higher intercept value. This also required a change to the main text (Page 6, paragraph 3). The main conclusions of this paragraph do not change.

Reviewer's comment 2.3

2. Page 13 line 2: Fig. 6e should be Fig. 7e.

Our response

The change was made as indicated.

Reviewer's comment 2.4

3. This is totally a personal request, I'd really like to hear some discussion on the cross-species prevalence of the non-autonomous/autonomous/semi-autonomous co-evolving system. Is it specific to wheat and barley? Does it have anything to do with the extreme genome size of wheat and barley?

Our response

This is a highly interesting question to which, at the moment, we have no definitive answer. Highly abundant non-autonomous elements are found in almost all organism groups (e.g. Alu in human, MITEs in all grasses, etc.). However, such a complex system of non-autonomous/autonomous/semi-autonomous co-evolving elements was new to us. The fact that we found (although very small numbers of) *RLG_WHAM* homologs in *Brachypodium* suggests that such systems can prevail also in small genomes. We have added a few statements discussing this point to the discussion (Page 15, paragraph 1).

Reviewer: 3

Comments to the Author

Wicker et al. present a detailed population-level survey of the highly abundant LTR retrotransposon families of the wheat genome. This approach is novel, and to my knowledge hasn't been applied with genome-scale data. Perhaps most interesting is the ability to track introgression from distinct lineages using TEs active at different times. As noted in the conclusion, the paper is of most interest to a specialist in wheat and transposable elements, due to the level of detail presented. It's almost a "natural history" of the wheat genome! Stylistically, there's a lot of interpretation in the results. Not sure on the journal requirements, but if possible merge results and discussion into one section? Below are a few major overarching comments, then comments that somewhat follow the narrative of the paper (I felt this was best because there unfortunately aren't line numbers on the manuscript I read).

Reviewer's comment 3.1

The github repository for the paper is empty, so I had a really hard time assessing how the variants were generated. After aligning individual copies to the consensus, what happens to indels? Do these come out in your VCF? If there's an insertion of 4 bp in one individual and an insertion of 2 bp in another individual, how are these integrated in the VCF? What about if there's an insertion of 4 bp in one sample and an insertion of 4 bp in another sample with a SNP variant within the insertion? If it's the case that indels are ignored, how do you end up with frameshifts?

Our response

We apologize for the empty github repository. This was an oversight. We had all scripts ready, but simply forgot to deposit them. All scripts and necessary instruction are now deposited at <https://github.com/wicker314/>.

The script that generates the vcf file compiles information from pairwise alignments of individual retrotransposon copies to a consensus sequence. Insertions in the individual retrotransposon are ignored, and deletions treated as missing data. This information was added to the methods section (Page 17, paragraph 3, see also comment 2.2 of reviewer 2).

Analyses of coding sequences (CDS) were done separately and for each copy individually by aligning it with a consensus of the predicted CDS. Here, frame shifts were spotted if the translated protein has regions with numerous stop codons. For molecular dating we simply selected those copies that encode a protein with no more than 3 stop codons, which makes it extremely unlikely that it contains a frame shift. We have added this information to the methods section (Page 17, paragraph 2).

Reviewer's comment 3.2

Throughout the paper, "wheat haplotype" is used to describe what I would call genotypes or cultivars. It's a little jarring to hear that a TE is active in a particular haplotype, since that's not really what's happening. To me, activity is jumps occurring within the nucleus of a germ cell of an individual. I'm not familiar enough with wheat nomenclature to know if this is a specialized term used to define something other than a contiguous section of sequence not disrupted by recombination (what I'd call a haplotype), but I'd suggest changing to genotype or cultivar for a general audience.

Our response

The reviewer's definition of haplotype is correct. This term is often (and incorrectly) used as "lab slang" in the wheat community, which is why it was used so prevalent in our manuscript. We changed it throughout the manuscript to more appropriate terms such as "wheat lineages", "wheat lines", "cultivars", "evolutionary lineages" or "sub-genome relatives", where necessary.

Reviewer's comment 3.3

Background, page 4 middle: It's written that "The LTRs are typically 1-2 kb long," referring to all plant LTR retrotransposons - that's not true of many plant species, where sometimes a majority of LTRs are <1000 bp, e.g. Figure 5 in Jedlicka, Lexa, and Kejnovsky (2020) *Frontiers in Plant Science* <https://www.frontiersin.org/articles/10.3389/fpls.2020.00644/full#F5>

Our response

The reviewer is correct. Even though in wheat, LTRs are usually longer, most are between 500 and 2000 bp. We made the statement more specific, mentioning LTRs are typically 0.5-2 kb in wheat and other Triticeae (Page 3, paragraph 2).

Reviewer's comment 3.4

The length of the TE identified is confusing to me, e.g. page 6 where "LTRs were found by chance in the same orientation and at the right distance." What does the right distance mean? In the methods, when ranges are provided, they're not centered on the consensus distance (e.g. 7800-9300, but the consensus is 8700 bp, so 900 bp shorter and 600 bp longer?). Are these the approximate lengths remaining after chopping 3% outlier tails of length?

Our response

We have made the statement more precise stating that the LTRs have to be approximately at the distance that is expected for the respective family (Page 5, paragraph 2). As for the size range: it was an oversight that this part of the methods describes a pilot study in barley (hence the example of RLC_BARE1). We have replaced it with the example of RLC_Angela where we indeed chose a size range of 1 kb centered on the consensus sequence (Page 16, paragraph 1).

Reviewer's comment 3.5

I disagree with the last paragraph of the dead on arrival section on page 7. The argument is made that low GC content means A/T rich stop codons are emerging through mutations more commonly. Since transition mutations are more common than transversion mutations, as well as the high rate of deamination of methylated cytosines to T, I think high GC content would make stop codons more likely to emerge. Can the authors describe their logic here?

Our response

The reviewer is correct. Our argument is one-sided, as it only considers sequence composition and does not consider transition-to-transversion ratio. We see that the statement is too speculative in this form and have removed it (Page 6, paragraph 3).

Reviewer's comment 3.6

When barley is introduced on page 8, it would be helpful to know how diverged in Mya wheat and barley are, to understand how the BARE1 homology can be expected.

Our response

Maybe the reviewer overlooked this, but it is actually stated that barley and wheat diverged about 8-10 million years ago (Page 7, paragraph 4).

Reviewer's comment 3.7

Page 10: When the G genome is introduced, I thought it was a typo. How G relates to A, B, and D needs to be introduced for a non-wheat audience.

Our response

We added a short explanation that *T. timopheevii* is a tetraploid that formed independently and combined species containing the sub-genomes A and G (page 9, paragraph 3).

Reviewer's comment 3.8

On the enrichment in introgressions, I buy the argument, but the pattern could also be generated by a different process. I assume regions of introgression are in regions with lower recombination, and thus lower selection efficacy. If TEs are effectively removed elsewhere in the genome, this pattern might be expected.

Our response

We did not find an enrichment of introgression in regions of low recombination. In fact, roughly half are at the ends of chromosomes (see Walkowiak et al., 2020, Supplementary Tables 13–16). Although theoretically, some TEs could be removed

selectively, we are not aware of any molecular mechanism that could selectively remove specific TE sub-families from the genome.

Reviewer's comment 3.9

With the Sabrina section, the internal proteins need more explanation. Even looking at figure 5, it's unclear what these short proteins are. What is a VLP gene? Is it GAG since that's generating the capsid? Do these proteins blast to genes of non-TE origin, or are they all deletion derivatives of Sabrina?

Our response

We have added more detailed information that two of the predicted proteins have strong 3D similarity to capsid and structural virus proteins (Page 11, paragraph 1) and provided more detail in the figure legend. We also added the reference to the prediction software to the methods (page 18, paragraph 3).

Reviewer's comment 3.10

Why do wheat homologs have a different name for some families but not all (BAGY2/Wilma being different vs. Sabrina or WHAM which seem to be the same in barley and wheat)? Is there higher sequence divergence of the autonomous copies than the nonautonomous? Although the BAGY2 presence in barley is described in the introduction, by the time I got to the results I had forgotten, and it's worth reiterating there.

Our response

We understand the reviewer's irritation. The naming of TE families is inconsistent for purely historical reasons. Some TE families were separately identified (and named) in wheat and barley, while others were named consistently in the context of comparative analyses. To clarify this, we have reiterated the naming of barley and wheat homologs as suggested (page 11, last 3).

Reviewer's comment 3.11

The five criteria presented for autonomous/nonautonomous families could be defined better. The first, "PCAs of autonomous and nonautonomous families should look similar" is tricky because what does "look similar" really mean? To me, a real test would be whether frameshift or premature stop codons are loaded on the PC that differentiates autonomous from nonautonomous families. That would be an actual falsifiable criteria for suggesting these are really coevolving, and also comparable across different families (the proportion of all frameshifts/stops that are loaded on the PC separating autonomous from nonautonomous). Also, the third requirement that the PBS needs the same tRNA primer doesn't seem true - is there evidence that RT needs anything other than a free 3' RNA end to get going? I get that there's a lot of stasis in PBS, as there's lots of nucleotides to change to evolve away to a brand new tRNA, but I don't think it's necessarily a requirement for autonomous/nonautonomous families.

Our response

We have added more detail on what we mean with "look similar" and refer to the example in Fig. 6 (page 11, paragraph 3). We understand that this sounds somewhat arbitrary, but providing a visual and intuitive representation of the data is the most powerful aspect of PCA, and particularly useful in the presented analyses. As to the issue of stop codons. For both, the RLC_Angela and the Wilma/WHAM/Sabrina system, the non-autonomous elements are so far diverged from the autonomous ones that CDS can simply not aligned in any meaningful way anymore. In fact, loading RLG_Wilma, WHAM and Sabrina into the same PCA would not work due to the lack of sequence conservation. Concerning the PBS, the reviewer is right that some retrotransposons (e.g. LINEs and SINEs) only require a free 3' end for replication. However, to our knowledge, LTR retrotransposons strictly require a tRNA primer to initiate reverse transcription. If the reviewer has a different insight, we would be very eager to learn about it and we would be happy to revise the manuscript accordingly.

Reviewer's comment 3.12

In the methods: More information about things like blast parameters, how the range of lengths was determined (it's not symmetric around the consensus length), what the expected polyprotein sequence is, are needed for reproducibility.

Our response

We have added the information to the methods section on parameters for blast searches, pairwise and multiple alignments and the production of the variant call file (pages 16-18, see also our response to comment 2.1).

Reviewer's comment 3.13

I was excited to see it written that the scripts for this study were available on github, but the repository is empty. It would have been a whole lot easier as a reviewer if they were available!

Our response

We apologize for the empty github repository. This was an oversight. We had all scripts ready, but simply forgot to deposit them. All scripts and necessary instruction are now deposited at <https://github.com/wicker314/>.

Reviewer's comment 3.14

Figures:

PC percentages loaded on x and y would be helpful - if something like 95% of the variation is explained by the first two PCs, it would make it clear whether there's more separation of "sub-populations" on PC3 that would separate them out to looking like "sub-families".

Our response

We have added to all PCA panels the percentage of variation explained by each of the principal components. Generally, the first two PCs explain roughly 40-60% of the variation, indicating that using the first two PCs is sufficiently robust to define sub-families. We also state this rationale now in the text (page 5, paragraph 4)

Reviewer's comment 3.15

Minor comments (sorry, no line numbers on the proof!):

Abstract: Third to last paragraph Tes isn't capitalized properly

Results: Page 6, principle should be principal in PCA

Figure 5 legend: typo, Poteinase

Supp Fig 3 needs a legend for what bubble size corresponds to what # of elements

Our response

All corrections and changes were made as indicated.

2 nd Editorial Decision	03-Sep-2021
-------------

The revised manuscript has addressed the comments of the reviewers and incorporated the recommendations of the editors, and we have now decided to accept the revised manuscript without further peer review.